# An Alternative to Variance: Gini Deviation for Risk-averse Policy Gradient

**Yudong Luo**[1,4], **Guiliang Liu**[2], **Pascal Poupart**[1,4], **Yangchen Pan**[3]
[1]University of Waterloo, [2]The Chinese University of Hong Kong, Shenzhen,
[3]University of Oxford, [4]Vector Institute
yudong.luo@uwaterloo.ca, liuguiliang@cuhk.edu.cn,
ppoupart@uwaterloo.ca, yangchen.pan@eng.ox.ac.uk

## Abstract

Restricting the variance of a policy's return is a popular choice in risk-averse Reinforcement Learning (RL) due to its clear mathematical definition and easy interpretability. Traditional methods directly restrict the total return variance. Recent methods restrict the per-step reward variance as a proxy. We thoroughly examine the limitations of these variance-based methods, such as sensitivity to numerical scale and hindering of policy learning, and propose to use an alternative risk measure, Gini deviation, as a substitute. We study various properties of this new risk measure and derive a policy gradient algorithm to minimize it. Empirical evaluation in domains where risk-aversion can be clearly defined, shows that our algorithm can mitigate the limitations of variance-based risk measures and achieves high return with low risk in terms of variance and Gini deviation when others fail to learn a reasonable policy.

## 1 Introduction

The demand for avoiding risks in practical applications has inspired risk-averse reinforcement learning (RARL). For example, we want to avoid collisions in autonomous driving [1], or avoid huge financial losses in portfolio management [2]. In addition to conventional RL, which finds policies to maximize the expected return [3], RARL also considers the control of risk.

Many risk measures have been studied for RARL, for instance, exponential utility functions [4], value at risk (VaR) [5], conditional value at risk (CVaR) [6, 7], and variance [8, 9]. In this paper, we mainly focus on the variance-related risk measures given their popularity, as variance has advantages in interpretability and computation [10, 11]. Such a paradigm is referred to as mean-variance RL. Traditional mean-variance RL methods consider the variance of the total return random variable. Usually, the total return variance is treated as a constraint to the RL problem, i.e., it is lower than some threshold [8, 9, 12]. Recently, [13] proposed a reward-volatility risk measure, which considers the variance of the per-step reward random variable. [13] shows that the per-step reward variance is an upper bound of the total return variance and can better capture the short-term risk. [14] further simplifies [13]'s method by introducing Fenchel duality.

Directly optimizing total return variance is challenging. It either necessitates double sampling [8] or calls for other techniques to avoid double sampling for faster learning [8, 9, 12]. As for the reward-volatility risk measure, [13] uses a complicated trust region optimization due to the modified reward's policy-dependent issue. [14] overcomes this issue by modifying the reward according to Fenchel duality. However, this reward modification strategy can possibly hinder policy learning by changing a "good" reward to a "bad" one, which we discuss in detail in this work.

To overcome the limitations of variance-based risk measures, we propose to use a new risk measure: Gini deviation (GD). We first review the background of mean-variance RL. Particularly, we explain the limitations of both total return variance and per-step reward variance risk measures. We then introduce

37th Conference on Neural Information Processing Systems (NeurIPS 2023).

GD as a dispersion measure for random variables and highlight its properties for utilizing it as a risk measure in policy gradient methods. Since computing the gradient using the original definition of GD is challenging, we derive the policy gradient algorithm from its quantile representation to minimize it. To demonstrate the effectiveness of our method in overcoming the limitations of variance-based risk measures, we modify several domains (Guarded Maze [7], Lunar Lander [15], Mujoco [16]) where risk-aversion can be clearly verified. We show that our method can learn risk-averse policy with high return and low risk in terms of variance and GD, when others fail to learn a reasonable policy.

## 2 Mean-Variance Reinforcement Learning

In standard RL settings, agent-environment interactions are modeled as a Markov decision process (MDP), represented as a tuple $(\mathcal{S}, \mathcal{A}, R, P, \mu_0, \gamma)$ [17]. $\mathcal{S}$ and $\mathcal{A}$ denote state and action spaces. $P(\cdot|s, a)$ defines the transition. $R$ is the state and action dependent reward variable, $\mu_0$ is the initial state distribution, and $\gamma \in (0, 1]$ is the discount factor. An agent follows its policy $\pi : \mathcal{S} \times \mathcal{A} \to [0, +\infty)$. The return at time step $t$ is defined as $G_t = \sum_{i=0}^{\infty} \gamma^i R(S_{t+i}, A_{t+i})$. Thus, $G_0$ is the random variable indicating the total return starting from the initial state following $\pi$.

Mean-variance RL aims to maximize $\mathbb{E}[G_0]$ and additionally minimize its variance $\mathbb{V}[G_0]$[8, 9, 12]. Generally, there are two ways to define a variance-based risk. The first one defines the variance based on the Monte Carlo **total return** $G_0$. The second defines the variance on the **per-step reward** $R$. We review these methods and their limitations in the following subsections. We will refer to $\pi$, $\pi_\theta$ and $\theta$ interchangeably throughout the paper when the context is clear.

### 2.1 Total Return Variance

Methods proposed by [8, 9, 12] consider the problem

$$\max_{\pi} \mathbb{E}[G_0], \ \text{s.t. } \mathbb{V}[G_0] \le \xi \tag{1}$$

where $\xi$ indicates the user's tolerance of the variance. Using the Lagrangian relaxation procedure [18], we can transform it to the following unconstrained optimization problem: $\max_{\pi} \mathbb{E}[G_0] - \lambda \mathbb{V}[G_0]$, where $\lambda$ is a trade-off hyper-parameter. Note that the mean-variance objective is in general NP-hard [19] to optimize. The main reason is that although variance satisfies a Bellman equation, it lacks the monotonicity of dynamic programming [20].

**Double Sampling in total return variance.** We first show how to solve unconstrained mean-variance RL via vanilla stochastic gradient. Suppose the policy is parameterized by $\theta$, define $J(\theta) = \mathbb{E}_{\pi}[G_0]$ and $M(\theta) := \mathbb{E}_{\pi}\big[(\sum_{t=0}^{\infty} \gamma^t R(S_t, A_t))^2\big]$, then $\mathbb{V}[G_0] = M(\theta) - J^2(\theta)$. The unconstrained mean-variance objective is equivalent to $J_\lambda(\theta) = J(\theta) - \lambda\big(M(\theta) - J^2(\theta)\big)$, whose gradient is

$$\nabla_\theta J_\lambda(\theta_t) = \nabla_\theta J(\theta_t) - \lambda \nabla_\theta(M(\theta) - J^2(\theta)) \tag{2}$$

$$= \nabla_\theta J(\theta_t) - \lambda\big(\nabla_\theta M(\theta) - 2J(\theta)\nabla_\theta J(\theta)\big) \tag{3}$$

The unbiased estimates for $\nabla_\theta J(\theta)$ and $\nabla_\theta M(\theta)$ can be estimated by approximating the expectations over trajectories by using a single set of trajectories, i.e., $\nabla_\theta J(\theta) = \mathbb{E}_\tau[R_\tau \omega_\tau(\theta)]$ and $\nabla_\theta M(\theta) = \mathbb{E}_\tau[R_\tau^2 \omega_\tau(\theta)]$, where $R_\tau$ is the return of trajectory $\tau$ and $\omega_\tau(\theta) = \sum_t \nabla_\theta \log \pi_\theta(a_t|s_t)$. In contrast, computing an unbiased estimate for $J(\theta)\nabla_\theta J(\theta)$ requires two distinct sets of trajectories to estimate $J(\theta)$ and $\nabla_\theta J(\theta)$ separately, which is known as double sampling.

**Remark.** Some work claims that double sampling cannot be implemented without having access to a generative model of the environment that allows users to sample at least two next states [12]. This is, however, not an issue in our setting where we allow sampling multiple trajectories. As long as we get enough trajectories, estimating $J(\theta)\nabla_\theta J(\theta)$ is possible.

Still, different methods were proposed to avoid this double sampling for faster learning. Specifically, [8] considers the setting $\gamma = 1$ and considers an unconstrained problem: $\max_\theta L_1(\theta) = \mathbb{E}[G_0] - \lambda g\big(\mathbb{V}[G_0] - \xi\big)$, where $\lambda > 0$ is a tunable hyper-parameter, and penalty function $g(x) = (\max\{0, x\})^2$. This method produces faster estimates for $\mathbb{E}[G_0]$ and $\mathbb{V}[G_0]$ and a slower updating for $\theta$ at each episode, which yields a two-time scale algorithm. [9] considers the setting $\gamma < 1$ and converts Formula 1 into an unconstrained saddle-point problem: $\max_\lambda \min_\theta L_2(\theta, \lambda) = -\mathbb{E}[G_0] + \lambda\big(\mathbb{V}[G_0] - \xi\big)$, where $\lambda$ is the dual variable. This approach uses perturbation method and smoothed function method to compute the gradient of value functions with respect to policy parameters. [12] considers the setting $\gamma = 1$, and introduces Fenchel duality $x^2 = \max_y(2xy - y^2)$ to avoid the term $J(\theta)\nabla_\theta J(\theta)$ in the gradient. The original problem is then transformed into $\max_{\theta, y} L_3(\theta, y) = 2y\big(\mathbb{E}[G_0] + \frac{1}{2\lambda}\big) - y^2 - \mathbb{E}[G_0^2]$, where $y$ is the dual variable.

**Limitations of Total Return Variance.** The presence of the square term $R_\tau^2$ in the mean-variance gradient $\nabla_\theta M(\theta) = \mathbb{E}_\tau[R_\tau^2 \omega_\tau(\theta)]$ (Equation 2) makes the gradient estimate sensitive to the numerical scale of the return, as empirically verified later. This issue is inherent in all methods that require computing $\nabla_\theta \mathbb{E}[G_0^2]$. Users can not simply scale the reward by a small factor to reduce the magnitude of $R_\tau^2$, since when scaling reward by a factor $c$, $\mathbb{E}[G_0]$ is scaled by $c$ but $\mathbb{V}[G_0]$ is scaled by $c^2$. Consequently, scaling the reward may lead to different optimal policies being obtained.

## 2.2 Per-step Reward Variance

A recent perspective uses per-step reward variance $\mathbb{V}[R]$ as a proxy for $\mathbb{V}[G_0]$. The probability mass function of $R$ is $\Pr(R = x) = \sum_{s,a} d_\pi(s,a) \mathbb{I}_{r(s,a)=x}$, where $\mathbb{I}$ is the indicator function, and $d_\pi(s,a) = (1-\gamma) \sum_{t=0}^\infty \gamma^t \Pr(S_t = s, A_t = a | \pi, P)$ is the normalized discounted state-action distribution. Then we have $\mathbb{E}[R] = (1-\gamma)\mathbb{E}[G_0]$ and $\mathbb{V}[G_0] \le \frac{\mathbb{V}[R]}{(1-\gamma)^2}$ (see Lemma 1 of [13]). Thus, [13] considers the following objective

$$\hat{J}_\lambda(\pi) = \mathbb{E}[R] - \lambda \mathbb{V}[R] = \mathbb{E}[R - \lambda(R - \mathbb{E}[R])^2] \tag{4}$$

This objective can be cast as a risk-neutral problem in the original MDP, but with a new reward function $\hat{r}(s,a) = r(s,a) - \lambda\big(r(s,a) - (1-\gamma)\mathbb{E}[G_0]\big)^2$. However, this $\hat{r}(s,a)$ is nonstationary (policy-dependent) due to the occurrence of $\mathbb{E}[G_0]$, so standard risk-neutral RL algorithms cannot be directly applied. Instead, this method uses trust region optimization [21] to solve.

[14] introduces Fenchel duality to Equation 4. The transformed objective is $\hat{J}_\lambda(\pi) = \mathbb{E}[R] - \lambda\mathbb{E}[R^2] + \lambda \max_y (2\mathbb{E}[R]y - y^2)$, which equals to $\max_{\pi,y} J_\lambda(\pi, y) = \sum_{s,a} d_\pi(s,a)\big(r(s,a) - \lambda r(s,a)^2 + 2\lambda r(s,a)y\big) - \lambda y^2$. The dual variable $y$ and policy $\pi$ are updated iteratively. In each inner loop $k$, $y$ has analytical solution $y_{k+1} = \sum_{s,a} d_{\pi_k}(s,a)r(s,a) = (1-\gamma)\mathbb{E}_{\pi_k}[G_0]$ since it is quadratic for $y$. After $y$ is updated, learning $\pi$ is a risk-neutral problem in the original MDP, but with a new modified reward

$$\hat{r}(s,a) = r(s,a) - \lambda r(s,a)^2 + 2\lambda r(s,a)y_{k+1} \tag{5}$$

Since $\hat{r}(s,a)$ is now stationary, any risk-neutral RL algorithms can be applied for policy updating.

**Limitations of Per-step Reward Variance. 1)** $\mathbb{V}[R]$ **is not an appropriate surrogate for** $\mathbb{V}[G_0]$ **due to fundamentally different implications.** Consider a simple example. Suppose the policy, the transition dynamics and the rewards are all deterministic, then $\mathbb{V}[G_0] = 0$ while $\mathbb{V}[R]$ is usually nonzero unless all the per-step rewards are equal. In this case, shifting a specific step reward by a constant will not affect $\mathbb{V}[G_0]$ and should not alter the optimal risk-averse policy. However, such shift can lead to a big difference for $\mathbb{V}[R]$ and may result in an invalid policy as we demonstrated in later example. **2) Reward modification hinders policy learning.** Since the reward modifications in [13] (Equation 4) and [14] (Equation 5) share the same issue, here we take Equation 5 as an example. This modification is likely to convert a positive reward to a much smaller or even negative value due to the square term, i.e. $-\lambda r(s,a)^2$. In addition, at the beginning of the learning phase, when the policy performance is not good, $y$ is likely to be negative in some environments (since $y$ relates to $\mathbb{E}[G_0]$). Thus, the third term $2\lambda r(s,a)y$ decreases the reward value even more. This prevents the agent to visit the good (i.e., rewarding) state even if that state does not contribute any risk. These two limitations raise a great challenge to subtly choose the value for $\lambda$ and design the reward for the environment.

**Empirical demonstration of the limitations.** Consider a maze problem (a modified version of Guarded Maze [7]) in Figure 1. Starting from the bottom left corner, the agent aims to reach the green goal state. The gray color corresponds to walls. The rewards for all states are deterministic (i.e., $-1$) except for the red state whose reward is a categorical distribution with mean $-1$. The reward for visiting the goal is a positive constant value. To reach the goal, a risk-neutral agent prefers the path at the bottom that goes through the red state, but $\mathbb{V}[G_0]$ will be nonzero. A risk-averse agent prefers the white path in the figure even though $\mathbb{E}[G_0]$ is slightly lower, but $\mathbb{V}[G_0] = 0$. Per-step reward variance methods aim to use $\mathbb{V}[R]$ as a proxy of $\mathbb{V}[G_0]$. For the risk-averse policy leading to the white path, ideally, increasing the goal reward by a constant will not effect $\mathbb{V}[G_0]$, but will make a big difference to $\mathbb{V}[R]$. For instance, when the goal reward is 10, $\mathbb{V}[R] = 10$. When goal reward is 20, $\mathbb{V}[R] \approx 36.4$, which is much more risk-averse. Next, consider the reward modification (Equation 5) for the goal

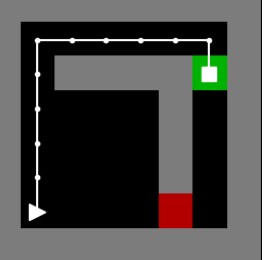

Figure 1: A modified Guarded Maze [7]. Red state returns an uncertain reward (details in text).

reward when it is 20. The square term in Equation 5 is $-400\lambda$. It is very easy to make the goal reward negative even for small $\lambda$, e.g., $0.1$. We do find this reward modification prevents the agent from reaching the goal in our experiments.

## 3 Gini Deviation as an Alternative of Variance

To avoid the limitations of $\mathbb{V}[G_0]$ and $\mathbb{V}[R]$ we have discussed, in this paper, we propose to use Gini deviation as an alternative of variance. Also, since GD has a similar definition and similar properties as variance, it serves as a more reasonable proxy of $\mathbb{V}[G_0]$ compared to $\mathbb{V}[R]$.

### 3.1 Gini Deviation: Definition and Properties

GD [22], also known as Gini mean difference or mean absolute difference, is defined as follows. For a random variable $X$, let $X_1$ and $X_2$ be two i.i.d. copies of $X$, i.e., $X_1$ and $X_2$ are independent and follow the same distribution as $X$. Then GD is given by

$$\mathbb{D}[X] = \frac{1}{2}\mathbb{E}[|X_1 - X_2|] \tag{6}$$

Variance can be defined in a similar way as $\mathbb{V}[X] = \frac{1}{2}\mathbb{E}[(X_1 - X_2)^2]$.

Given samples $\{x_i^1\}_{i=1}^n$ from $X_1$ and $\{x_j^2\}_{j=1}^n$ from $X_2$. The unbiased empirical estimations for GD and variance are $\hat{\mathbb{D}}[X] = \frac{1}{2n^2}\sum_{i=1}^n\sum_{j=1}^n|x_i^1 - x_j^2|$ and $\hat{\mathbb{V}}[X] = \frac{1}{2n^2}\sum_{i=1}^n\sum_{j=1}^n(x_i^1 - x_j^2)^2$.

Both risk profiles aim to measure the variability of a random variable and share similar properties [23]. For example, they are both location invariant, and can be presented as a weighted sum of order statistics. [23] argues that the GD is superior to the variance as a measure of variability for distributions far from Gaussian. We refer readers to this paper for a full overview. Here we highlight two properties of $\mathbb{D}[X]$ to help interpret it. Let $\mathcal{M}$ denote the set of real random variables and let $\mathcal{M}^p, p \in [1, \infty)$ denote the set of random variables whose probability measures have finite $p$-th moment, then

- $\mathbb{V}[X] \geq \sqrt{3}\,\mathbb{D}[X]$ for all $X \in \mathcal{M}^2$.
- $\mathbb{D}[cX] = c\mathbb{D}[X]$ for all $c > 0$ and $X \in \mathcal{M}$.

The first property is known as Glasser's inequality [24], which shows $\mathbb{D}[X]$ is a lower bound of $\mathbb{V}[X]$ if $X$ has finite second moment. The second one is known as positive homogeneity in coherent measures of variability [25], and is also clear from the definition of GD in Equation 6. In RL, considering $X$ is the return variable, this means GD is less sensitive to the reward scale compared to variance, i.e., scaling the return will scale $\mathbb{D}[X]$ linearly, but quadratically for $\mathbb{V}[X]$. We also provide an intuition of the relation between GD and variance from the perspective of convex order, as shown in Appendix 7. Note also that while variance and GD are both measures of variability, GD is a *coherent* measure of variability [25]. Appendix 12 provides a discussion of the properties of coherent measures of *variability*, while explaining the differences with coherent measures of *risk* such as conditional value at risk (CVaR).

### 3.2 Signed Choquet Integral for Gini Deviation

This section introduces the concept of signed Choquet integral, which provides an alternative definition of GD and makes gradient-based optimization convenient. Note that with the original definition (Equation 6), it can be intractable to compute the gradient w.r.t. the parameters of a random variable's density function through its GD.

The Choquet integral [26] was first used in statistical mechanics and potential theory and was later applied to decision making as a way of measuring the expected utility [27]. The signed Choquet integral belongs to the Choquet integral family and is defined as:

**Definition 1** ([28], Equation 1). *A signed Choquet integral* $\Phi_h : X \to \mathbb{R}, X \in \mathcal{L}^\infty$ *is defined as*

$$\Phi_h(X) = \int_{-\infty}^0 \Big(h\big(\Pr(X \geq x)\big) - h(1)\Big)dx + \int_0^\infty h\big(\Pr(X \geq x)\big)dx \tag{7}$$

*where* $\mathcal{L}^\infty$ *is the set of bounded random variables in a probability space, $h$ is the distortion function and $h \in \mathcal{H}$ such that $\mathcal{H} = \{h : [0, 1] \to \mathbb{R}, h(0) = 0, h$ is of bounded variation$\}$.*

This integral has become the building block of law-invariant risk measures [1] after the work of [29, 30]. One reason for why signed Choquet integral is of interest to the risk research community is that it

---

[1]Law-invariant property is one of the popular "financially reasonable" axioms. If a functional returns the same value for two random variables with the same distribution, then the functional is called law-invariant.

is not necessarily monotone. Since most practical measures of variability are not monotone, e.g., variance, standard deviation, or deviation measures in [31], it is possible to represent these measures in terms of $\Phi_h$ by choosing a specific distortion function $h$.

**Lemma 1** ([28], Section 2.6). *Gini deviation is a signed Choquet integral with a concave $h$ given by $h(\alpha) = -\alpha^2 + \alpha, \alpha \in [0, 1]$.*

This Lemma provides an alternative definition for GD, i.e., $\mathbb{D}[X] = \int_{-\infty}^{\infty} h\big(\Pr(X \geq x)\big)dx, h(\alpha) = -\alpha^2 + \alpha$. However, this integral is still not easy to compute. Here we turn to its quantile representation for easy calculation.

**Lemma 2** ([28], Lemma 3). *$\Phi_h(X)$ has a quantile representation. If $F_X^{-1}$ is continuous, then $\Phi_h(X) = \int_1^0 F_X^{-1}(1 - \alpha)dh(\alpha)$, where $F_X^{-1}$ is the quantile function (inverse CDF) of X.*

Combining Lemma 1 and 2, $\mathbb{D}[X]$ can be computed alternatively as

$$\mathbb{D}[X] = \Phi_h(X) = \int_0^1 F_X^{-1}(1 - \alpha)dh(\alpha) = \int_0^1 F_X^{-1}(\alpha)(2\alpha - 1)d\alpha \tag{8}$$

With this quantile representation of GD, we can derive a policy gradient method for our new learning problem in the next section. It should be noted that variance cannot be directly defined by a $\Phi_h$-like quantile representation, but as a complicated related representation: $\mathbb{V}[X] = \sup_{h \in \mathcal{H}} \big\{ \Phi_h(X) - \frac{1}{4}\|h'\|_2^2 \big\}$, where $\|h'\|_2^2 = \int_0^1 (h'(p))^2 dp$ if $h$ is continuous, and $\|h'\|_2^2 := \infty$ if it is not continuous (see Example 2.2 of [32]). Hence, such representation of the conventional variance measure is not readily usable for optimization.

## 4 Policy Gradient for Mean-Gini Deviation

In this section, we consider a new learning problem by replacing the variance with GD. Specifically, we consider the following objective

$$\max_{\pi} \mathbb{E}[G_0] - \lambda\mathbb{D}[G_0] \tag{9}$$

where $\lambda$ is the trade-off parameter. To maximize this objective, we may update the policy towards the gradient ascent direction. Computing the gradient for the first term has been widely studied in risk-neutral RL [3]. Computing the gradient for the second term may be difficult at the first glance from its original definition, however, it becomes possible via its quantile representation (Equation 8).

### 4.1 Gini Deviation Gradient Formula

We first give a general gradient calculation for GD of a random variable $Z$, whose distribution function is parameterized by $\theta$. In RL, we can interpret $\theta$ as the policy parameters, and $Z$ as the return under that policy, i.e., $G_0$. Denote the Probability Density Function (PDF) of $Z$ as $f_Z(z; \theta)$. Given a confidence level $\alpha \in (0, 1)$, the $\alpha$-level quantile of $Z$ is denoted as $q_\alpha(Z; \theta)$, and given by

$$q_\alpha(Z; \theta) = F_{Z_\theta}^{-1}(\alpha) = \inf\big\{z : \Pr(Z_\theta \leq z) \geq \alpha\big\} \tag{10}$$

For technical convenience, we make the following assumptions, which are also realistic in RL.

**Assumption 1.** *$Z$ is a continuous random variable, and bounded in range $[-b, b]$ for all $\theta$.*

**Assumption 2.** *$\frac{\partial}{\partial \theta_i} q_\alpha(Z; \theta)$ exists and is bounded for all $\theta$, where $\theta_i$ is the i-th element of $\theta$.*

**Assumption 3.** *$\frac{\partial f_Z(z; \theta)}{\partial \theta_i} / f_Z(z; \theta)$ exists and is bounded for all $\theta, z$. $\theta_i$ is the i-th element of $\theta$.*

Since $Z$ is continuous, the second assumption is satisfied whenever $\frac{\partial}{\partial \theta_i} f_Z(z; \theta)$ is bounded. These assumptions are common in likelihood-ratio methods, e.g., see [33]. Relaxing these assumptions is possible but would complicate the presentation.

**Proposition 1.** *Let Assumptions 1, 2, 3 hold. Then*

$$\nabla_\theta \mathbb{D}[Z_\theta] = -\mathbb{E}_{z \sim Z_\theta}\Big[\nabla_\theta \log f_Z(z; \theta)\int_z^b \big(2F_{Z_\theta}(t) - 1\big)dt\Big] \tag{11}$$

*Proof.* By Equation 8, the gradient of $\mathbb{D}[Z_\theta] = \Phi_h(Z_\theta)$ $(h(\alpha) = -\alpha^2 + \alpha, \alpha \in [0, 1])$ is

$$\nabla_\theta \mathbb{D}[Z_\theta] = \nabla_\theta \Phi_h(Z_\theta) = \int_0^1 (2\alpha - 1)\nabla_\theta F_{Z_\theta}^{-1}(\alpha)d\alpha = \int_0^1 (2\alpha - 1)\nabla_\theta q_\alpha(Z; \theta)d\alpha. \tag{12}$$

This requires to calculate the gradient for any $\alpha$-level quantile of $Z_\theta$, i.e., $\nabla_\theta q_\alpha(Z; \theta)$. Based on the

assumptions and the definition of the $\alpha$-level quantile, we have $\int_{-b}^{q_\alpha(Z;\theta)} f_Z(z;\theta)dz = \alpha$. Taking a derivative and using the Leibniz rule we obtain

$$0 = \nabla_\theta \int_{-b}^{q_\alpha(Z;\theta)} f_Z(z;\theta)dz = \int_{-b}^{q_\alpha(Z;\theta)} \nabla_\theta f_Z(z;\theta)dz + \nabla_\theta q_\alpha(Z;\theta) f_Z\big(q_\alpha(Z;\theta);\theta\big) \qquad (13)$$

Rearranging the term, we get $\nabla_\theta q_\alpha(Z;\theta) = -\int_{-b}^{q_\alpha(Z;\theta)} \nabla_\theta f_Z(z;\theta)dz \cdot \big[f_Z\big(q_\alpha(Z;\theta);\theta\big)\big]^{-1}$. Plugging back to Equation 12 gives us an intermediate version of $\nabla_\theta \mathbb{D}[Z_\theta]$.

$$\nabla_\theta \mathbb{D}[Z_\theta] = -\int_0^1 (2\alpha - 1) \int_{-b}^{q_\alpha(Z;\theta)} \nabla_\theta f_Z(z;\theta)dz \cdot \big[f_Z\big(q_\alpha(Z;\theta);\theta\big)\big]^{-1} d\alpha \qquad (14)$$

By switching the integral order of Equation 14 and applying $\nabla_\theta \log(x) = \frac{1}{x}\nabla_\theta x$, we get the final gradient formula Equation 11. The full calculation is in Appendix 8.1.

## 4.2 Gini Deviation Policy Gradient via Sampling

In a typical application, $Z$ in Section 4.1 would correspond to the performance of a system, e.g., the total return $G_0$ in RL. Note that in order to compute Equation 11, one needs access to $\nabla_\theta \log f_Z(z;\theta)$: the sensitivity of the system performance to the parameters $\theta$. Usually, the system performance is a complicated function and calculating its probability distribution is intractable. However, in RL, the performance is a function of trajectories. The sensitivity of the trajectory distribution is often easy to compute. This naturally suggests a sampling based algorithm for gradient estimation.

Now consider Equation 11 in the context of RL, i.e., $Z = G_0$ and $\theta$ is the policy parameter.

$$\nabla_\theta \mathbb{D}[G_0] = -\mathbb{E}_{g \sim G_0}\big[\nabla_\theta \log f_{G_0}(g;\theta) \int_g^b \big(2F_{G_0}(t) - 1\big)dt\big] \qquad (15)$$

To sample from the total return variable $G_0$, we need to sample a trajectory $\tau$ from the environment by executing $\pi_\theta$ and then compute its corresponding return $R_\tau := r_1 + \gamma r_2 + ... + \gamma^{T-1}r_T$, where $r_t$ is the per-step reward at time $t$, and $T$ is the trajectory length. The probability of the sampled return can be calculated as $f_{G_0}(R_\tau;\theta) = \mu_0(s_0)\prod_{t=0}^{T-1}[\pi_\theta(a_t|s_t)p(r_{t+1}|s_t,a_t)]$. The gradient of its log-likelihood is the same as that of $P(\tau|\theta) = \mu_0(s_0)\prod_{t=0}^{T-1}[\pi_\theta(a_t|s_t)p(s_{t+1}|s_t,a_t)]$, since the difference in transition probability does not alter the policy gradient. It is well known that $\nabla_\theta \log P(\tau|\theta) = \sum_{t=0}^{T-1} \nabla_\theta \log \pi_\theta(a_t|s_t)$.

For the integral part of Equation 15, it requires the knowledge of the CDF of $G_0$. In practice, this means we should obtain the full value distribution of $G_0$, which is usually not easy. One common approach to acquire an empirical CDF or quantile function (inverse CDF) is to get the quantile samples of a distribution and then apply some reparameterization mechanism. For instance, reparameterization is widely used in distributional RL for quantile function estimation. The quantile function has been parameterized as a step function [34, 35], a piece-wise linear function [36], or other higher order spline functions [37]. In this paper, we use the step function parameterization given its simplicity. To do so, suppose we have $n$ trajectory samples $\{\tau_i\}_{i=1}^n$ from the environment and their corresponding returns $\{R_{\tau_i}\}_{i=1}^n$, the returns are sorted in ascending order such that $R_{\tau_1} \leq R_{\tau_2} \leq ... \leq R_{\tau_n}$, then each $R_{\tau_i}$ is regarded as a quantile value of $G_0$ corresponding to the quantile level $\alpha_i = \frac{1}{2}(\frac{i-1}{n} + \frac{i}{n})$, i.e., we assume $q_{\alpha_i}(G_0;\theta) = R_{\tau_i}$. This strategy is also common in distributional RL, e.g., see [34, 38]. The largest return $R_{\tau_n}$ is regarded as the upper bound $b$ in Equation 15.

Thus, given ordered trajectory samples $\{\tau_i\}_{i=1}^n$, an empirical estimation for GD policy gradient is (a detailed example is given in Appendix 8.2)

$$-\frac{1}{n-1}\sum_{i=1}^{n-1} \eta_i \sum_{t=0}^{T-1} \nabla_\theta \log \pi_\theta(a_{i,t}|s_{i,t}), \text{ where } \eta_i = \sum_{j=i}^{n-1} \frac{2j}{n}\big(R_{\tau_{j+1}} - R_{\tau_j}\big) - \big(R_{\tau_n} - R_{\tau_i}\big) \qquad (16)$$

The sampled trajectories can be used to estimate the gradient for $\mathbb{E}[G_0]$ in the meantime, e.g., the well known vanilla policy gradient (VPG), which has the form $\mathbb{E}_\tau[R_\tau \sum_{t=0}^{T-1} \nabla_\theta \log \pi_\theta(a_t|s_t)]$. It is more often used as $\mathbb{E}_\tau[\sum_{t=0}^{T-1} \nabla_\theta \log \pi_\theta(a_t|s_t)\gamma^t g_t]$, where $g_t = \sum_{t'=t}^{T-1} \gamma^{t'-t}r(s_{t'},a_{t'})$, which is known to have lower variance. Usually $g_t$ is further subtracted by a value function to improve stability, called REINFORCE with baseline. Apart from VPG, another choice to maximize $\mathbb{E}[G_0]$ is using PPO [39].

## 4.3 Incorporating Importance Sampling

For on-policy policy gradient, samples are abandoned once the policy is updated, which is expensive for our gradient calculation since we are required to sample $n$ trajectories each time. To improve

the sample efficiency to a certain degree, we incorporate importance sampling (IS) to reuse samples for multiple updates in each loop. For each $\tau_i$, the IS ratio is $\rho_i = \prod_{t=0}^{T-1} \pi_\theta(a_{i,t}|s_{i,t})/\pi_{\hat{\theta}}(a_{i,t}|s_{i,t})$, where $\hat{\theta}$ is the old policy parameter when $\{\tau_i\}_{i=1}^n$ are sampled. Suppose the policy gradient for maximizing $\mathbb{E}[G_0]$ is REINFORCE baseline. With IS, the empirical mean-GD policy gradient is

$$\frac{1}{n}\sum_{i=1}^n \rho_i \sum_{t=0}^{T-1} \nabla_\theta \log \pi_\theta(a_{i,t}|s_{i,t})(g_{i,t} - V(s_{i,t})) + \frac{\lambda}{n-1}\sum_{i=1}^{n-1} \rho_i \eta_i \sum_{t=0}^{T-1} \nabla_\theta \log \pi_\theta(a_{i,t}|s_{i,t}) \quad (17)$$

where $g_{i,t}$ is the sum of rewards-to-go as defined above. $V(s_{i,t})$ is the value function. The first part can also be replaced by PPO-Clip policy gradient. Then we have

$$\frac{1}{n}\sum_{i=1}^n \sum_{t=0}^{T-1} \nabla_\theta \min\big(\frac{\pi_\theta(a_{i,t}|s_{i,t})}{\pi_{\hat{\theta}}(a_{i,t}|s_{i,t})}A_{i,t}, f(\epsilon, A_{i,t})\big) + \frac{\lambda}{n-1}\sum_{i=1}^{n-1} \rho_i \eta_i \sum_{t=0}^{T-1} \nabla_\theta \log \pi_\theta(a_{i,t}|s_{i,t}) \quad (18)$$

where $A_{i,t}$ is the advantage estimate, and $f()$ is the clip function in PPO with $\epsilon$ being the clip range, i.e. $f(\epsilon, A_{i,t}) = \text{clip}(\frac{\pi_\theta(a_{i,t}|s_{i,t})}{\pi_{\hat{\theta}}(a_{i,t}|s_{i,t})}, 1 - \epsilon, 1 + \epsilon)A_{i,t}$.

The extreme IS values $\rho_i$ will introduce high variance to the policy gradient. To stabilize learning, one strategy is that in each training loop, we only select $\tau_i$ whose $\rho_i$ lies in $[1 - \delta, 1 + \delta]$, where $\delta$ controls the range. The updating is terminated if the chosen sample size is lower than some threshold, e.g., $\beta \cdot n, \beta \in (0, 1)$. Another strategy is to directly clip $\rho_i$ by a constant value $\zeta$, i.e., $\rho_i = \min(\rho_i, \zeta)$, e.g., see [40]. In our experiments, we use the first strategy for Equation 17, and the second for Equation 18. We leave other techniques for variance reduction of IS for future study. The full algorithm that combines GD with REINFORCE and PPO is in Appendix 9.

## 5 Experiments

Our experiments were designed to serve two main purposes. First, we investigate whether the GD policy gradient approach could successfully discover risk-averse policies in scenarios where variance-based methods tend to fail. To accomplish this, we manipulated reward choices to assess the ability of the GD policy gradient to navigate risk-averse behavior. Second, we sought to verify the effectiveness of our algorithm in identifying risk-averse policies that have practical significance in both discrete and continuous domains. We aimed to demonstrate its ability to generate meaningful risk-averse policies that are applicable and valuable in practical settings.

**Baselines.** We compare our method with the original mean-variance policy gradient (Equation 2, denoted as MVO), Tamar's method [8] (denoted as Tamar), MVP [12], and MVPI [14]. Specifically, MVO requires multiple trajectories to compute $J(\theta)\nabla_\theta J(\theta)$. We use $\frac{n}{2}$ trajectories to estimate $J(\theta)$ and another $\frac{n}{2}$ to estimate $\nabla_\theta J(\theta)$, where $n$ is the sample size. MVPI is a general framework for policy iteration whose inner risk-neutral RL solver is not specified. For the environment with discrete actions, we build MVPI on top of Q-Learning or DQN [41]. For continuous action environments, MVPI is built on top of TD3 [42] as in [14]. We use REINFORCE to represent the REINFORCE with baseline method. We use MG as a shorthand of mean-GD to represent our method. In each domain, we ensure each method's policy or value nets have the same neural network architecture.

For policy updating, MVO and MG collect $n$ episodes before updating the policy. In contrast, Tamar and MVP update the policy after each episode. Non-tabular MVPI updates the policy at each environment step. In hyperparameter search, we use the parameter search range in MVPI [14] as a reference, making reasonable refinements to find an optimal parameter setting. Please refer to Appendix 10 for any missing implementation details. Code is available at[2].

### 5.1 Tabular case: Modified Guarded Maze Problem

This domain is a modified Guarded Maze [7] that was previously described in Section 2.2. The original Guarded Maze is asymmetric with two openings to reach the top path (in contrast to a single opening for the bottom path). In addition, paths via the top tend to be longer than paths via the bottom. We modified the maze to be more symmetric in order to reduce preferences arising from certain exploration strategies that might be biased towards shorter paths or greater openings, which may confound risk aversion. Every movement before reaching the goal receives a reward of $-1$ except moving to the red state, where the reward is sampled from $\{-15, -1, 13\}$ with probability $\{0.4, 0.2, 0.4\}$ (mean is $-1$) respectively. The maximum episode length is 100. MVO and MG collect $n = 50$ episodes before updating the policy. Agents are tested for 10 episodes per evaluation.

---

[2]https://github.com/miyunluo/mean-gini

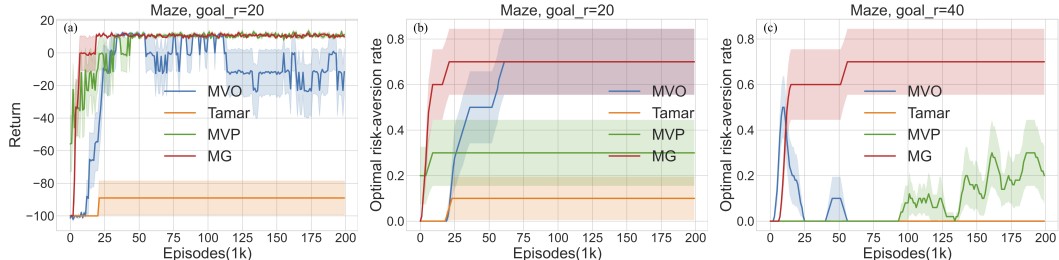

Figure 2: (a) Policy evaluation return and (b,c) optimal risk-aversion rate v.s. training episodes in Maze. Curves are averaged over 10 seeds with shaded regions indicating standard errors. For optimal risk-aversion rate, higher is better.

**The failure of variance-based baselines under simple reward manipulation.** We first set the goal reward to 20. Here, we report the optimal risk-aversion rate achieved during training. Specifically, we measure the percentage of episodes that obtained the optimal risk-averse path, represented by the white color path in Figure 1, out of all completed episodes up to the current stage of training.

Notice that MVO performs well in this domain when using double sampling to estimate its gradient. Then we increase the goal reward to 40. This manipulation does not affect the return variance of the optimal risk-averse policy, since the reward is deterministic. However, the performances of MVO, Tamar, MVP all decrease, since they are more sensitive to the numerical scale of the return (due to the $\mathbb{E}[G_0^2]$ term introduced by variance). MVPI is a policy iteration method in this problem, whose learning curve is not intuitive to show. It finds the optimal risk-averse path when the goal reward is 20, but it fails when the goal reward is 40. An analysis for MVPI is given in Appendix 10.2.2. We compare the sensitivity of different methods with respect to $\lambda$ in Appendix 10.2.4.

**Remark.** Scaling rewards by a small factor is not an appropriate approach to make algorithms less sensitive to the numerical scale for both total return variance and per-step reward variance, since it changes the original mean-variance objective in both cases.

### 5.2 Discrete control: LunarLander

This domain is taken from OpenAI Gym Box2D environments [15]. We refer readers to its official documents for the full description. Originally, the agent is awarded 100 if it comes to rest. We divide the ground into two parts by the middle line of the landing pad, as shown in Figure 10 in Appendix. If the agent lands in the right area, an additional noisy reward sampled from $\mathcal{N}(0, 1)$ times 90 is given. A risk-averse agent should learn to land at the left side as much as possible. We include REINFORCE as a baseline to demonstrate the risk-aversion of our algorithm. REINFORCE, MVO and MG collect $n = 30$ episodes before updating their policies. Agents are tested for 10 episodes per evaluation.

We report the rate at which different methods land on the left in Figure 3(b) (we omit the failed methods), i.e, the percentage of episodes successfully landing on the left per evaluation. MVO, Tamar, and MVP do not learn reasonable policies in this domain according to their performances in Figure 3(a). MVP learns to land in the middle of the learning phase, but soon after fails to land. Since successfully landing results in a large return (success reward is 100), the return square term ($\mathbb{E}[G_0^2]$) introduced by variance makes MVP unstable. MVPI also fails to land since $\mathbb{V}[R]$ is sensitive to the numerical scale of rewards. In this domain, the success reward is much larger than other reward values. Furthermore, reward modification in MVPI turns large success rewards into negative values, which prevents the agent from landing on the ground. MG achieves a comparable return with REINFORCE, but clearly learns a risk-averse policy by landing more on the left.

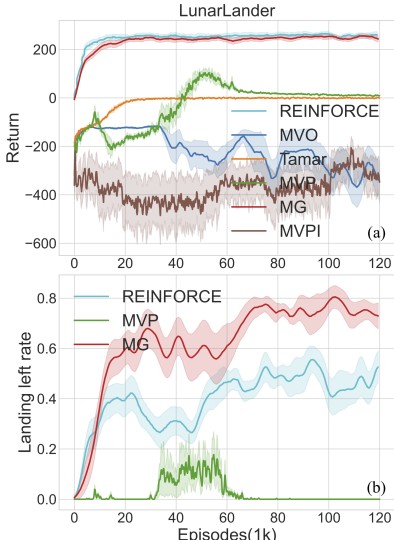

Figure 3: (a) Policy evaluation return and (b) left-landing rate (i.e., risk-averse landing rate) v.s. training episodes in LunarLander. Curves are averaged over 10 seeds with shaded regions indicating standard errors. For landing left rate, higher is better.

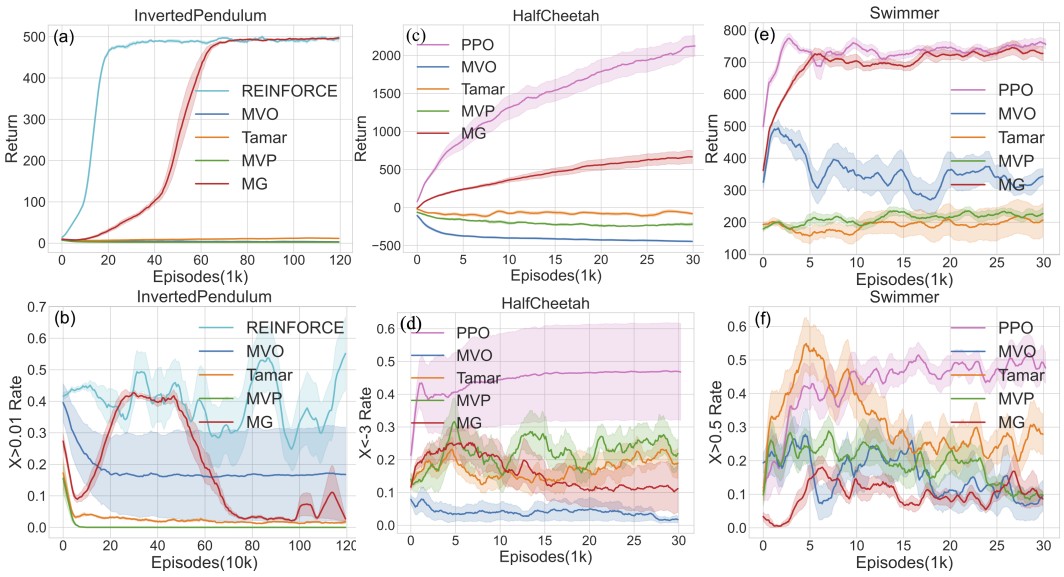

Figure 4: (a,c,e) Policy evaluation return and (b,d,f) location visiting rate v.s. training episodes in Mujoco of episode-based methods. Curves are averaged over 10 seeds with shaded regions indicating standard errors. For location visiting rate, lower is better.

## 5.3 Continuous control: Mujoco

Mujoco [16] is a collection of robotics environments with continuous states and actions in OpenAI Gym [15]. Here, we selected three domains (InvertedPendulum, HalfCheetah, and Swimmer) that are conveniently modifiable, where we are free to modify the rewards to construct risky regions in the environment (Through empirical testing, risk-neutral learning failed when similar noise was introduced to other Mujoco domains. Consequently, identifying the cause for the failure of risk-averse algorithms on other domains became challenging). Motivated by and following [43, 44], we define a risky region based on the X-position. For instance, if X-position $> 0.01$ in InvertedPendulum, X-position $< -3$ in HalfCheetah, and X-position $> 0.5$ in Swimmer, an additional noisy reward sampled from $\mathcal{N}(0, 1)$ times 10 is given. Location information is appended to the agent's observation. A risk-averse agent should reduce the time it visits the noisy region in an episode. We also include the risk-neutral algorithms as baselines to highlight the risk-aversion degree of different methods.

All the risk-averse policy gradient algorithms still use VPG to maximize the expected return in InvertedPendulum (thus the risk-neutral baseline is REINFORCE). Using VPG is also how these methods are originally derived. However, VPG is not good at more complex Mujoco domains, e.g., see [45]. In HalfCheetah and Swimmer, we combine those algorithms with PPO-style policy gradient to maximize the expected return. Minimizing the risk term remains the same as their original forms. MVPI is an off-policy time-difference method in Mujoco. We train it with 1e6 steps instead of as many episodes as other methods. MVO and MG sample $n = 30$ episodes in InvertedPendulum and $n = 10$ in HalfCheetah and Swimmer before updating policies. Agents are tested for 20 episodes per evaluation. The percentage of time steps visiting the noisy region in an episode is shown in Figure 4(b,d,f). Compared with other return variance methods, MG achieves a higher return while maintaining a lower visiting rate. Comparing MVPI and TD3 against episode-based algorithms like MG is not straightforward within the same figure due to the difference in parameter update frequency. MVPI and TD3 update parameters at each environment time step. We shown their learning curves in Figure 5. MVPI also learns risk-averse policies in all three domains according to its learning curves.

We further design two domains using HalfCheetah and Swimmer. The randomness of the noisy reward linearly decreases when agent's forward distance grows. To maximize the expected return and minimize risk, the agent has to move forward as far as possible. The results are shown in Figures 18,19 in Appendix. In these two cases, only MG shows a clear tendency of moving forward, which suggests our method is less sensitive to reward choices compared with methods using $\mathbb{V}[R]$.

The return variance and GD during learning in the above environments are also reported in Appendix 10. In general, when other return variance based methods can find the risk-averse policy, MG maintains a lower or comparable return randomness when measured by both variance and GD. When

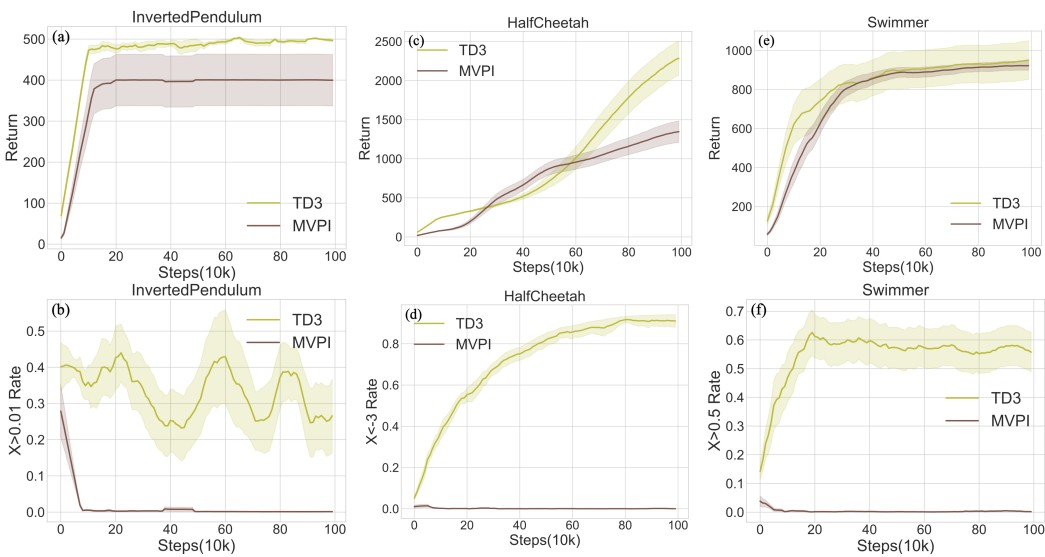

Figure 5: (a,c,e) Policy evaluation return and (b,d,f) location visiting rate v.s. training episodes in Mujoco of TD3 and MVPI. Curves are averaged over 10 seeds with shaded regions indicating standard errors. For location visiting rate, lower is better.

other methods fail to learn a reasonably good risk-averse policy, MG consistently finds a notably higher return and lower risk policy compared with risk-neutral methods. MVPI has the advantage to achieve low return randomness in location based risky domains, since minimizing $\mathbb{V}[R]$ naturally avoids the agent from visiting the noisy region. But it fails in distance-based risky domains.

## 6 Conclusion and Future Work

This paper proposes to use a new risk measure, Gini deviation, as a substitute for variance in mean-variance RL. It is motivated to overcome the limitations of the existing total return variance and per-step reward variance methods, e.g., sensitivity to numerical scale and hindering of policy learning. A gradient formula is presented and a sampling-based policy gradient estimator is proposed to minimize such risk. We empirically show that our method can succeed when the variance-based methods will fail to learn a risk-averse or a reasonable policy. This new risk measure may inspire a new line of research in RARL. First, one may study the practical impact of using GD and variance risk measures. Second, hybrid risk-measure may be adopted in real-world applications to leverage the advantages of various risk measures.

**Limitations and future work.** Our mean-GD policy gradient requires sampling multiple trajectories for one parameter update, making it less sample efficient compared to algorithms that can perform updates per environment step or per episode. As a result, one potential avenue for future work is to enhance the sample efficiency of our algorithm. This can be achieved by more effectively utilizing off-policy data or by adapting the algorithm to be compatible with online, incremental learning.

## Acknowledgments and Disclosure of Funding

We thank Ruodu Wang from University of Waterloo and Han Wang from University of Alberta for valuable discussions and insights. Resources used in this work were provided, in part, by the Province of Ontario, the Government of Canada through CIFAR, companies sponsoring the Vector Institute `https://vectorinstitute.ai/partners/` and the Natural Sciences and Engineering Council of Canada. Yudong Luo is also supported by a David R. Cheriton Graduate Scholarship, a President's Graduate Scholarship, and an Ontario Graduate Scholarship. Guiliang Liu's research was in part supported by the Start-up Fund UDF01002911 of the Chinese University of Hong Kong, Shenzhen. Yangchen Pan acknowledges funding from the Turing AI World Leading Fellow.

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

## Supplementary Information

## 7 Convex Order, Gini Deviation, and Variance

Convex order describes dominance in terms of variability and is widely used in actuarial science.

**Definition 2** ([46])**.** *Consider two random variables $X$ and $Y$, $X$ is called convex order smaller than $Y$, succinctly $X \leq_{cx} Y$, if $\mathbb{E}[\psi(X)] \leq \mathbb{E}[\psi(Y)]$, for all convex function $\psi()$, assuming that both expectations exist.*

In convex order $X \leq_{cx} Y$, $Y$ is also called a mean-preserving spread of $X$ [47], which intuitively means that $Y$ is more spread-out (and hence more random) than $X$. Thus, it is often desirable for a measure of variability to be monotone with respect to convex order [25]. Both variance and GD, as a measure of variability, are consistent with convex order, i.e.,

- If $X \leq_{cx} Y$, then $\mathbb{V}[X] \leq \mathbb{V}[Y]$ for all $X, Y \in \mathcal{M}$
- If $X \leq_{cx} Y$, then $\mathbb{D}[X] \leq \mathbb{D}[Y]$ for all $X, Y \in \mathcal{M}$

*Proof.* It is immediate that $X \leq_{cx} Y$ implies $\mathbb{E}[X] = \mathbb{E}[Y]$. If we take convex function $\psi(x) = x^2$, we can get the order of variance $\mathbb{V}[X] \leq \mathbb{V}[Y]$. For the proof of GD, please refer to the following Lemma. Recall that GD can be expressed in the form of signed Choquet integral with a concave function $h$.

**Lemma 3** ([28],Theorem 2)**.** *Convex order consistency of a signed Choquet integral is equivalent to its distortion function $h$ being concave, i.e., $X \leq_{cx} Y$ if and only if the signed Choquet integral $\Phi_h(X) \leq \Phi_h(Y)$ for all concave functions $h \in \mathcal{H}$.*

## 8 GD Gradient Formula Calculation

### 8.1 General GD Gradient Formula

**Proposition 1.** *Let Assumptions 1, 2, 3 hold. Then*

$$\nabla_\theta \mathbb{D}[Z_\theta] = -\mathbb{E}_{z \sim Z_\theta} \left[ \nabla_\theta \log f_Z(z; \theta) \int_z^b \big( 2F_{Z_\theta}(t) - 1 \big) dt \right] \tag{11}$$

Consider a random variable $Z$, whose distribution function is controlled by $\theta$. Recall that $q_\alpha(Z; \theta)$ represents the $\alpha$-level quantile of $Z_\theta$. According to Equation 14, the gradient of GD is

$$\nabla_\theta \mathbb{D}[Z_\theta] = \nabla_\theta \Phi_h(Z_\theta) = -\int_0^1 (2\alpha - 1) \int_{-b}^{q_\alpha(Z; \theta)} \nabla_\theta f_Z(z; \theta) dz \frac{1}{f_Z\big(q_\alpha(Z; \theta); \theta\big)} d\alpha$$

To make the integral over $\alpha$ clearer, we rewrite $q_\alpha(Z; \theta)$ as $F_{Z_\theta}^{-1}(\alpha)$, where $F_{Z_\theta}$ is the CDF.

$$\nabla_\theta \mathbb{D}[Z_\theta] = -\int_0^1 (2\alpha - 1) \int_{-b}^{F_{Z_\theta}^{-1}(\alpha)} \nabla_\theta f_Z(z; \theta) dz \frac{1}{f_Z(F_{Z_\theta}^{-1}(\alpha); \theta)} d\alpha$$

Switching the integral order, we get

$$\begin{aligned}
\nabla_\theta \mathbb{D}[Z_\theta] &= -\int_{-b}^b \int_{F_{Z_\theta}(z)}^1 (2\alpha - 1) \nabla_\theta f_Z(z; \theta) \frac{1}{f_Z(F_{Z_\theta}^{-1}(\alpha); \theta)} d\alpha dz \\
&= -\int_{-b}^b \nabla_\theta f_Z(z; \theta) \int_{F_{Z_\theta}(z)}^1 (2\alpha - 1) \frac{1}{f_Z(F_{Z_\theta}^{-1}(\alpha); \theta)} d\alpha dz
\end{aligned} \tag{19}$$

Denote $t = F_{Z_\theta}^{-1}(\alpha)$, then $\alpha = F_{Z_\theta}(t)$. Here, we further change the inner integral from $d\alpha$ to $dF_{Z_\theta}(t)$, i.e., $d\alpha = dF_{Z_\theta}(t) = f_Z(t; \theta) dt$. The integral range for $t$ is now from $F_{Z_\theta}^{-1}(F_{Z_\theta}(z)) = z$ to $F_{Z_\theta}^{-1}(1) = b$.

$$\begin{aligned}
\nabla_\theta \mathbb{D}[Z_\theta] &= -\int_{-b}^b \nabla_\theta f_Z(z; \theta) \int_z^b \big( 2F_{Z_\theta}(t) - 1 \big) \frac{1}{f_Z(t; \theta)} dF_{Z_\theta}(t) \, dz \\
&= -\int_{-b}^b \nabla_\theta f_Z(z; \theta) \int_z^b \big( 2F_{Z_\theta}(t) - 1 \big) dt \, dz
\end{aligned} \tag{20}$$

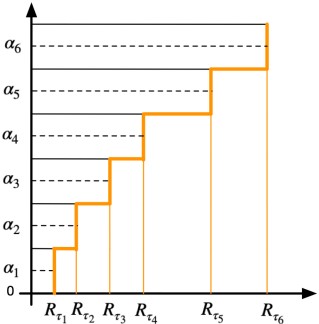

Figure 6: An example of parameterizing (inverse) CDF given six quantiles. The function is highlighted in the bold line of orange color.

Applying $\nabla_\theta \log(x) = \frac{1}{x} \nabla_\theta x$ to $\nabla_\theta f_Z(z; \theta)$, we have

$$
\begin{aligned}
\nabla_\theta \mathbb{D}[Z_\theta] &= -\int_{-b}^{b} f_Z(z; \theta) \nabla_\theta \log f_Z(z; \theta) \int_{z}^{b} \big(2F_{Z_\theta}(t) - 1\big) dt \, dz \\
&= -\mathbb{E}_{z \sim Z_\theta}\Big[\nabla_\theta \log f_Z(z; \theta) \int_{z}^{b} \big(2F_{Z_\theta}(t) - 1\big) dt\Big]
\end{aligned}
\tag{21}
$$

### 8.2  GD Policy Gradient via Sampling

Following Section 4.2, we consider estimating the following equation via sampling.

$$
\nabla_\theta \mathbb{D}[G_0] = -\mathbb{E}_{g \sim G_0}\Big[\nabla_\theta \log f_{G_0}(g; \theta) \int_{g}^{b} \big(2F_{G_0}(t) - 1\big) dt\Big]
$$

Here sampling from $G_0$ corresponds to sampling trajectory $\tau$ with its return $R_\tau$ from the environment. As discussed in the main paper, for a trajectory $\tau_i$, $\nabla_\theta \log f_{G_0}(R_{\tau_i}, \theta)$ is estimated as $\sum_{t=0}^{T-1} \nabla_\theta \log \pi_\theta(a_{i,t}|s_{i,t})$ .

The CDF function $F_{G_0}$ is parameterized by a step function given its quantiles $\{R_{\tau_i}\}_{i=1}^{n}$, which satisfy $R_{\tau_1} \le R_{\tau_2} \le ... \le R_{\tau_n}$. An example of the step function is shown in Figure 6. With this parameterization, the integral over CDF can be regarded as the area below the step function.

Thus, for each $\tau_i$, the integral over CDF is approximated as ($R_{\tau_n}$ is treated as $b$)

$$
\int_{R_{\tau_i}}^{R_{\tau_n}} 2F_{G_0}(t) dt \approx \sum_{j=i}^{n-1} 2 \times \frac{j}{n} \big(R(\tau_{j+1}) - R(\tau_j)\big)
\tag{22}
$$

Aggregating all the calculations together yields Equation 16.

## 9  Mean-GD Policy Gradient Algorithm

We consider maximizing $\mathbb{E}[G_0] - \lambda \mathbb{D}[G_0]$ in this paper. Maximizing the first term, i.e., $\mathbb{E}[G_0]$ has been widely studied in risk-neutral RL. For on-policy policy gradient, we can use vanilla policy gradient (VPG), e.g., REINFORCE with baseline, or more advanced techniques like PPO [39].

To make fair comparison with other risk-averse policy gradient methods, we first initialize mean-GD policy gradient with VPG. To improve sample efficiency, we incorporate IS for multiple updates. Taking advantage of having $n$ samples, we can select those trajectories whose IS ratio is in the range $[1 - \delta, 1 + \delta]$ for calculation to reduce gradient variance. In more complex continuous domains, e.g., Mujoco, we combine GD policy gradient with PPO since VPG is not good at Mujoco [45]. However, in this case, policy may have a significant difference per update, where the former IS selection strategy can no longer be applied. Then we directly clip the IS ratio by a constant, though it is biased [40]. The full algorithms are summarized in Algorithm 1 and 2. We omit the parameters for PPO in Algorithm 2 for simplicity. We still report the PPO parameter settings in Section 10.

---
**Algorithm 1** Mean-Gini Deviation Policy Gradient (with REINFORCE baseline)
---
**Input:** Iterations number $K$, sample size $n$, inner update number $M$, policy learning rate $\alpha_\theta$, value learning rate $\alpha_\phi$, importance sampling range $\delta$, inner termination parameter $\beta$, trade-off parameter $\lambda$.
Initialize policy $\pi_\theta$ parameter $\theta$, value $V_\phi$ parameter $\phi$.
**for** $k = 1$ **to** $K$ **do**
   Sample $n$ trajectories $\{\tau_i\}_{i=1}^n$ by $\pi_\theta$, compute return $\{R(\tau_i)\}_{i=1}^n$
   Compute rewards-to-go for each state in $\tau_i$: $g_{i,t}$
   **for** $m = 1$ **to** $M$ **do**
      Compute importance sampling ratio for each trajectory $\{\rho_i\}_{i=1}^n$
      Select $\mathcal{D} = \{\tau_s\}$ whose $\rho_s \in [1 - \delta, 1 + \delta]$
      Sort trajectories such that $R(\tau_1) \leq ... \leq R(\tau_{|\mathcal{D}|})$
      **if** $|\mathcal{D}| < n \cdot \beta$ **then**
         break
      **end if**
      mean_grad = 0, gini_grad = 0
      **for** $i = 1$ **to** $|\mathcal{D}|$ **do**
         mean_grad += $\rho_i \cdot \sum_0^{T-1} \nabla_\theta \log \pi_\theta(a_{i,t}|s_{i,t})(g_{i,t} - V\phi(s_{i,t}))$
         Update $V_\phi$ by mean-squared error $\frac{1}{T} \sum_{t=0}^{T-1} (V_\phi(s_{i,t}) - g_{i,t})^2$ with learning rate $\alpha_\phi$
      **end for**
      **for** $i = 1$ **to** $|\mathcal{D}| - 1$ **do**
         gini_grad += $-\rho_i \cdot \eta_i \sum_{t=0}^{T-1} \nabla_\theta \log \pi_\theta(a_{i,t}|s_{i,t})$, where
         $\eta_i = \sum_{j=i}^{|\mathcal{D}|-1} \frac{2j}{|\mathcal{D}|}(R_{j+1} - R_j) - (R_{|\mathcal{D}|} - R_i)$
      **end for**
      Update $\pi_\theta$ by $\left( \frac{1}{|\mathcal{D}|} \text{ mean\_grad} - \frac{\lambda}{|\mathcal{D}|-1} \text{ gini\_grad} \right)$ with learning rate $\alpha_\theta$ (Equation 17)
   **end for**
**end for**
---

## 10 Experiments Details

### 10.1 General Descriptions of Different Methods

Among the methods compared in this paper, Tamar [8] and MVP [12] are on-policy policy gradient methods. MVO and MG are policy gradient methods, but sample $n$ trajectories and use IS to update. Non-tabular MVPI [14] is an off-policy time-difference method.

**Policy gradeint methods.** MVO, Tamar, MVP, are originally derived based on VPG. Since VPG is known to have a poor performance in Mujoco, we also combined these mean-variance methods with PPO. Thus these mean-variance methods and our mean-GD method have different instantiations to maximize the expected return in different domains:

- With VPG: in Maze, LunarLander, InvertedPendulum.
- With PPO: in HalfCheetah, Swimmer.

When the risk-neutral policy gradient is VPG, for MVO and MG, it is REINFORCE with baseline; for Tamar and MVP, we strictly follow their papers to implement the algorithm, where no value function is used. MVO and MG collect $n$ trajectories and use the IS strategy in Algorithm 1 to update policies. Tamar and MVP do not need IS, and update policies at the end of each episode.

When the risk-neutral policy gradient is PPO, we augment PPO with the variance or GD policy gradient from the original methods. MVO and MG collect $n$ trajectories and use the IS strategy in Algorithm 2 to compute the gradient for the risk term. Tamar and MVP still update policies once at the end of each episode.

**MVPI.** We implemented three versions of MVPI in different domains:

- Tabular: in Maze, MVPI is a policy iteration method (Algorithm 1 in [14]).
- With DQN: in LunarLander, since this environment has discrete action space.
- With TD3: in InvertedPendulum, HalfCheetah, Swimmer, since these environments have continuous action space.

---
**Algorithm 2** Mean-Gini Deviation Policy Gradient (with PPO)
---

**Input:** Iterations number $K$, sample size $n$, inner update number $M$, policy learning rate $\alpha_\theta$, value learning rate $\alpha_\phi$, importance sampling clip bound $\zeta$, trade-off parameter $\lambda$.
Initialize policy $\pi_\theta$ parameter $\theta$, value $V_\phi$ parameter $\phi$.
**for** $k = 1$ **to** $K$ **do**
   Sample $n$ trajectories $\{\tau_i\}_{i=1}^n$ by $\pi_\theta$, compute return $\{R(\tau_i)\}_{i=1}^n$
   Compute rewards-to-go for each state in $\tau_i$: $g_{i,t}$
   Compute advantages for each state-action in $\tau_i$: $A(s_{i,t}, a_{i,t})$ based on current $V_\phi$
   **for** $m = 1$ **to** $M$ **do**
      Compute importance sampling ratio for each trajectory $\{\rho_i\}_{i=1}^n$, and $\rho_i = \min(\rho_i, b)$
      Sort trajectories such that $R(\tau_1) \leq ... \leq R(\tau_n)$
      mean_grad = 0, gini_grad = 0
      **for** $i = 1$ **to** $n$ **do**
         mean_grad += PPO-Clip actor grad
         Update $V_\phi$ by mean-squared error $\frac{1}{T}\sum_{t=0}^{T-1}(V_\phi(s_{i,t}) - g_{i,t})^2$ with learning rate $\alpha_\phi$
      **end for**
      **for** $i = 1$ **to** $n-1$ **do**
         gini_grad += $-\rho_i \cdot \eta_i \sum_{t=0}^{T-1} \nabla_\theta \log \pi_\theta(a_{i,t}|s_{i,t})$, where
         $\eta_i = \sum_{j=i}^{n-1} \frac{2j}{n}(R_{j+1} - R_j) - (R_n - R_i)$
      **end for**
      Update $\pi_\theta$ by $\left(\frac{1}{nT}\text{ mean\_grad} - \frac{\lambda}{(n-1)T}\text{ gini\_grad}\right)$ with learning rate $\alpha_\theta$ (Equation 18)
   **end for**
**end for**
---

We summarize the components required in different methods in Table 1.

Table 1: Model components in different methods

|  | Policy func | Value func | Additional training variables |
|---|:---:|:---:|:---:|
| MVO-VPG | √ | √ | |
| MVO-PPO | √ | √ | |
| Tamar-VPG | √ | × | J,V (mean,variance) |
| Tamar-PPO | √ | √ | J,V (mean,variance) |
| MVP-VPG | √ | × | y (dual variable) |
| MVP-PPO | √ | √ | y (dual variable) |
| MG-VPG | √ | √ | |
| MG-PPO | √ | √ | |
| MVPI-Q-Learning | × | √ | |
| MVPI-DQN | × | √ | |
| MVPI-TD3 | √ | √ | |

## 10.2 Modified Guarded Maze Problem

The maze consists of a $6 \times 6$ grid. The agent can visit every free cell without a wall. The agent can take four actions (up, down, left, right). The maximum episode length is 100.

**Policy function.** For methods requiring a policy function, i.e., MVO, Tamar, MVP, MG, the policy is represented as

$$\pi_\theta(a|s) = \frac{e^{\phi(s,a) \cdot \theta}}{\sum_b e^{\phi(s,b) \cdot \theta}} \tag{23}$$

where $\phi(s, a)$ is the state-action feature vector. Here we use one-hot encoding to represent $\phi(s, a)$. Thus, the dimension of $\phi(s, a)$ is $6 \times 6 \times 4$. The derivative of the logarithm is

$$\nabla_\theta \log \pi_\theta(a|s) = \phi(s, a) - \mathbb{E}_{b \sim \pi_\theta(\cdot|s)} \phi(s, b) \tag{24}$$

**Value function.** For methods requiring a value function, i.e., REINFORCE baseline used in MVO and MG, and Q-learning in MVPI, the value function is represented as $V_\omega(s) = \phi(s) \cdot \omega$ or $Q_\omega(s, a) = \phi(s, a) \cdot \omega$. Similarly, $\phi(s)$ is a one-hot encoding.

**Optimizer.** The policy and value loss are optimized by stochastic gradient descent (SGD).

### 10.2.1 Learning Parameters

We set discount factor $\gamma = 0.999$.

**MVO**: policy learning rate is 1e-5 $\in$ {5e-5, 1e-5, 5e-6}, value function learning rate is 100 times policy learning rate. $\lambda = 1.0 \in$ {0.6, 0.8, 1.0, 1.2}. Sample size $n = 50$. Maximum inner update number $M = 10$. IS ratio range $\delta = 0.5$. Inner termination ratio $\beta = 0.6$.

**Tamar**: policy learning rate is 1e-5 $\in$ {5e-5, 1e-5, 5e-6}, $J, V$ learning rate is 100 times policy learning rate. Threshold $b = 50 \in$ {10, 50, 100}, $\lambda = 0.1 \in$ {0.1, 0.2, 0.4}.

**MVP**: policy learning rate is 1e-5 $\in$ {5e-5, 1e-5, 5e-6}, $y$ learning rate is the same. $\lambda = 0.1 \in$ {0.1, 0.2, 0.4}.

**MG**: policy learning rate is 1e-4 $\in$ {5e-4, 1e-4, 5e-5}, value function learning rate is 100 times policy learning rate. $\lambda = 1.2 \in$ {0.8, 1.0, 1.2}. Sample size $n = 50$. Maximum inner update number $M = 10$. IS ratio range $\delta = 0.5$. Inner termination ratio $\beta = 0.6$.

**MVPI**: Q function learning rate 5e-3 $\in$ {5e-3, 1e-3, 5e-4}, $\lambda = 0.2 \in$ {0.2, 0.4, 0.6}.

### 10.2.2 Analysis for MVPI in Maze (MVPI-Q-Learning)

MVPI-Q-Learning finds the optimal risk-averse path when goal reward is 20 but fails when goal reward is 40. Since it is not intuitive to report the learning curve for a policy iteration method where its reward is modified in each iteration, we give an analysis here.

The value of dual variable $y$ in Equation 5 is $(1 - \gamma)\mathbb{E}[G_0]$ given the current policy. Recall that the maximum episode length is 100. At the beginning, when the Q function is randomly initialized (i.e., it is a random policy), $\mathbb{E}[G_0] = \sum_{t=0}^{99} 0.999^t(-1) \approx -95.2$. Thus $y = (1 - 0.999) \times (-95.2) = -0.0952$, the goal reward after modification is $r_{\text{goal}} = 20 - 0.2 \times 20^2 + 2 \times 0.2 \times 20 \times y \approx -60.7$. For the red state, its original reward is sampled from $\{-15, -1, 13\}$. After the reward modification, it becomes sampling from $\{-59.4, -1.16, -21.2\}$. Thus the expected reward of the red state is now $r_{\text{red}} = 0.4 \times (-59.4) + 0.2 \times (-1.16) + 0.4 \times (-21.2) = -32.472$. Given the maximum episode length is 100, the optimal policy is still the white path in Figure 1. (Because the expected return for the white path is $\sum_{t=0}^{9} 0.999^t(-1) + 0.999^{10}(-60.7) \approx -70$. The expected return for a random walk is $\sum_{t=0}^{99} 0.999^t(-1) \approx -95.2$. The expected return for the shortest path going through the red state is even lower than the white path since the reward of the red state after modification is pretty negative: $-32.472$.)

However, when goal reward is 40, after modification, the goal reward becomes $r_{\text{goal}} = 40 - 0.2 \times 40^2 + 2 \times 0.2 \times 40 \times y \approx -281.5$. In this case, the optimal policy has to avoid the goal state since it leads to a even lower return.

### 10.2.3 Return Variance and Gini Deviation in Maze

We report the return's variance and GD during learning for different methods, as shown in Figure 7 and 8. Tamar [8] is unable to reach the goal in both settings. MVO fails to reach the goal when the return magnitude increases. MVP [12]'s optimal risk-aversion rate is much lower than MG. MG can learn a risk averse policy in both settings with lower variance and GD, which suggests it is less sensitive to the return numerical scale.

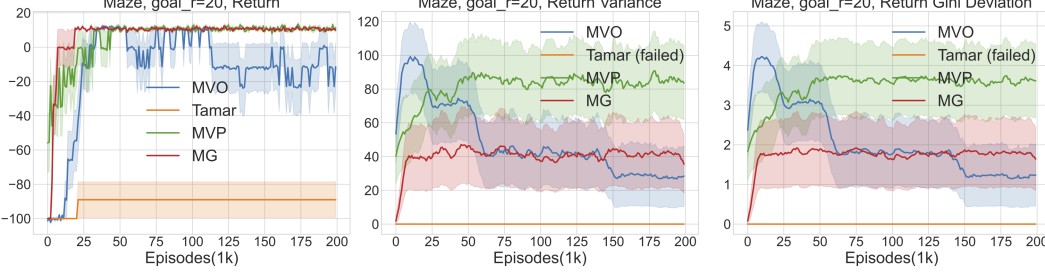

Figure 7: Expected return, return variance and Gini deviation of different methods in Maze when goal reward is 20. Curves are averaged over 10 seeds with shaded regions indicating standard errors.

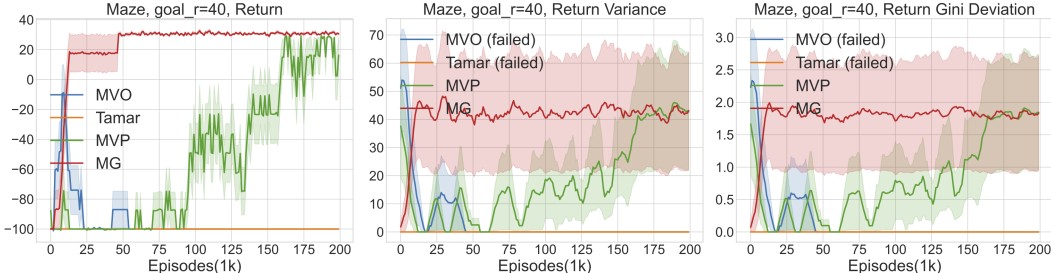

Figure 8: Expected return, return variance and Gini deviation of different methods in Maze when goal reward is 40. Curves are averaged over 10 seeds with shaded regions indicating standard errors.

### 10.2.4 Sensitivity to Trade-off Parameter

We report the learning curves of total return based methods with different $\lambda$ when goal reward is 20 in Figure 9. The learning parameters are the same as shown in Section 10.2.1.

## 10.3 LunarLander Discrete

The agent's goal is to land the lander on the ground without crashing. The state dimension is 8. The action dimension is 4. The detailed reward information is available at this webpage [3]. We divide the whole ground into left and right parts by the middle line of the landing pad as shown in Figure 10. If the agent lands in the right part, an additional noisy reward signal sampled from $\mathcal{N}(0,1)$ tims 90 is given. We set the maximum episode length to 1000. Note that the original reward for successfully landing is 100, thus the numerical scale of both return and reward is relatively large in this domain.

**Policy function.** The policy is a categorical distribution in REINFORCE, MVO, Tamar, MVP and MG, modeled as a neural network with two hidden layers. The hidden size is 128. Activation is ReLU. Softmax function is applied to the output to generate categorical probabilities.

**Value function.** The value function in REINFORCE, MVO, MG, and $Q$ function in MVPI-DQN is a neural network with two hidden layers. The hidden size is 128. Activation is ReLU.

**Optimizer.** The optimizer for policy and value functions is Adam.

### 10.3.1 Learning Parameters

Discount factor is $\gamma = 0.999$

**REINFORCE** (with baseline): the policy learning rate is 7e-4 $\in$ {7e-4, 3e-4, 7e-5}, value function learning rate is 10 times policy learning rate.

**MVO**: policy learning rate is 7e-5 $\in$ {7e-4, 3e-4, 7e-5}, value function learning rate is 10 times policy learning rate. $\lambda = 0.4 \in \{0.4, 0.6, 0.8\}$. Sample size $n = 30$. Maximum inner update number $M = 10$. IS ratio range $\delta = 0.5$. Inner termination ratio $\beta = 0.6$.

**Tamar**: policy learning rate is 7e-5 $\in$ {7e-4, 3e-4, 7e-5}. $J, V$ learning rate is 100 times the policy learning rate. Threshold $b = 50 \in \{10,50,100\}$. $\lambda = 0.2 \in \{0.2, 0.4, 0.6\}$.

**MVP**: policy learning rate is 7e-5 $\in$ {7e-4, 3e-4, 7e-5}. $y$ learning rate is the same. $\lambda = 0.2 \in \{0.2, 0.4, 0.6\}$.

**MG**: policy learning rate is 7e-4 $\in$ {7e-4, 3e-4, 7e-5}, value function learning rate is 10 times policy learning rate. $\lambda = 0.6 \in \{0.4, 0.6, 0.8\}$. Sample size $n = 30$. Maximum inner update number $M = 10$. IS ratio range $\delta = 0.5$. Inner termination ratio $\beta = 0.6$.

**MVPI**: Q function learning rate is 7e-4 $\in$ {7e-4, 3e-4, 7e-5}, $\lambda = 0.2 \in \{0.2, 0.4, 0.6\}$. Batch size is 64.

### 10.3.2 Return Variance and Gini Deviation in LunarLander

The return's variance and GD of different methods during training is shown in Figure 11. All the risk-averse methods, apart from ours, fail to learn a reasonable policy in this domain. Our method achieves a comparable return, but with lower variance and GD compared with risk-neutral method.

---

[3]https://www.gymlibrary.dev/environments/box2d/lunar_lander/

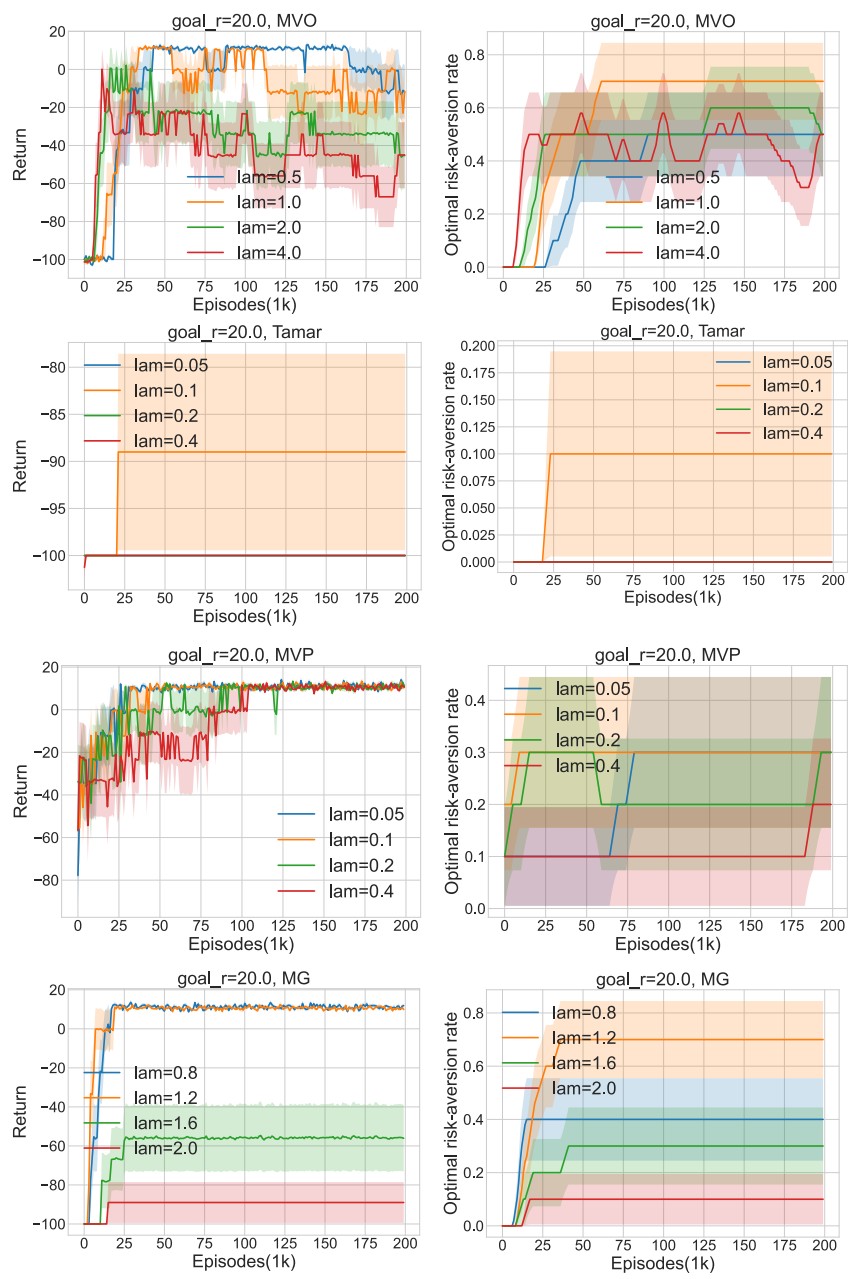

Figure 9: Policy evaluation return and optimal risk-aversion rate v.s. training episodes in Maze (goal reward is 20) for MVO, Tamar, MVP, and MG with different $\lambda$. The reasonable $\lambda$ range varies in different methods. Curves are averaged over 10 seeds with shaded regions indicating standard errors.

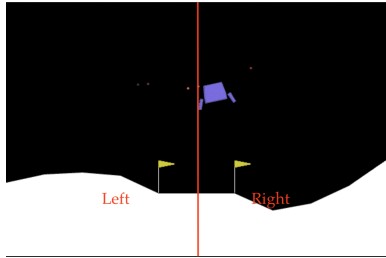

Figure 10: Divide the ground of LunarLander into left and right parts by the middle (red) line. If landing in the right area, an additional reward sampled from $\mathcal{N}(0,1)$ times 90 is given.

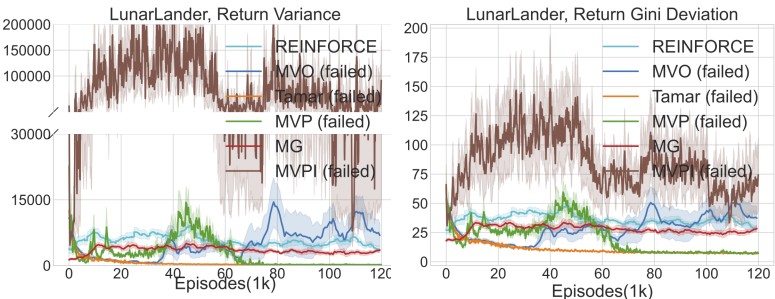

Figure 11: Return variance and Gini deviation of different methods in LunarLander. Curves are averaged over 10 seeds with shaded regions indicating standard errors.

## 10.4 InvertedPendulum

(The description of the Mujoco environments can be found at this webpage [4].)

The agent's goal is to balance a pole on a cart. The state dimension is $4$ (X-position is already contained). The action dimension is $1$. At each step, the environment provides a reward of $1$. If the agent reaches the region X-coordinate $> 0.01$, an additional noisy reward signal sampled from $\mathcal{N}(0,1)$ times 10 is given. To avoid the initial random speed forcing the agent to the X-coordinate $> 0.01$ region, we decrease the initial randomness for the speed from $U(-0.01, 0.01)$ to $U(-0.0005, 0.0005)$, where $U()$ represents the uniform distribution. The game ends if angle between the pole and the cart is greater than 0.2 radian or a maximum episode length 500 is reached.

**Policy function.** The policy is a normal distribution in REINFORCE, and VPG based methods (MVO, Tamar, MVP, and MG), modeled as a neural network with two hidden layers. The hidden size is 128. Activation is ReLU. Tanh is applied to the output to scale it to $(-1, 1)$. The output times the maximum absolute value of the action serves as the mean of the normal distribution. The logarithm of standard deviation is an independent trainable parameter.

The policy is a deterministic function in TD3 and MVPI, modeled as a neural network with two hidden layers. The hidden size is 128. Activation is ReLU. Tanh is applied to the output to scale it to $(-1, 1)$. The output times the maximum absolute value of the action is the true action executed in the environment.

**Value function.** The value function in REINFORCE, VPG based MVO, VPG based MG, TD3, and MVPI is a neural network with two hidden layers. The hidden size is 128. Activation is ReLU.

**Optimizer.** Optimizer for both policy and value function is Adam.

### 10.4.1 Learning Parameters

Discount factor $\gamma = 0.999$.

**REINFORCE** (with baseline): policy learning rate is 1e-4 $\in$ {1e-4, 5e-5, 5e-4}, value function learning rate is 10 times policy learning rate.

---

[4]https://www.gymlibrary.dev/environments/mujoco/

**MVO**: policy learning rate is 1e-5 $\in$ {1e-4, 5e-5, 1e-5}, value function learning rate is 10 times policy learning rate. $\lambda = 0.6 \in$ {0.2, 0.4, 0.6}. Sample size $n = 30$. Maximum inner update number $M = 10$. IS ratio range $\delta = 0.5$. Inner termination ratio $\beta = 0.6$.

**Tamar**: policy learning rate is 1e-5 $\in$ {1e-4, 5e-5, 1e-5}. $J, V$ learning rate is 100 times policy learning rate. Threshold $b = 50 \in$ {10,50,100}. $\lambda = 0.2 \in$ {0.2, 0.4, 0.6}.

**MVP**: policy learning rate is 1e-5 $\in$ {1e-4, 5e-4, 1e-5}. $y$ learning rate is the same. $\lambda = 0.2 \in$ {0.2, 0.4, 0.6}.

**MG**: policy learning rate is 1e-4 $\in$ {1e-4, 5e-5, 1e-4}, value function learning rate is 10 times policy learning rate. $\lambda = 1.0 \in$ {0.6, 1.0, 1.4}. Sample size $n = 30$. Maximum inner update number $M = 10$. IS ratio range $\delta = 0.5$. Inner termination ratio $\beta = 0.6$.

**MVPI**: Policy and value function learning rate 3e-4 $\in$ {3e-4, 7e-5, 1e-5}, $\lambda = 0.2 \in$ {0.2, 0.4, 0.6}. Batch size is 256.

**TD3**: Policy and value function learning rate 3e-4 $\in$ {3e-4, 7e-5, 1e-5}. Batch size is 256.

#### 10.4.2 Return Variance and Gini Deviation in InvertedPendulum

The return varaince and GD of different methods are shown in Figure 12 and 13.

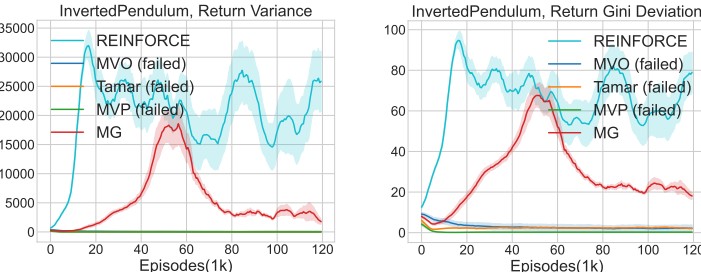

Figure 12: Return variance and Gini deviation of policy gradient methods in InvertedPendulum. Curves are averaged over 10 seeds with shaded regions indicating standard errors.

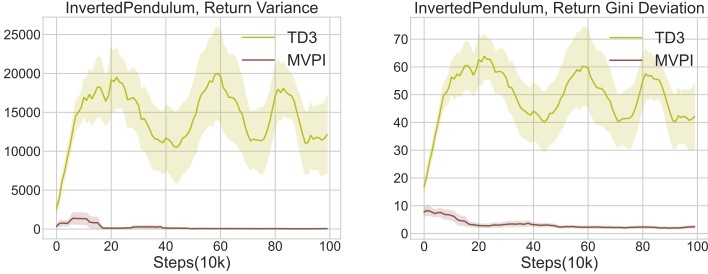

Figure 13: Return variance and Gini deviation of TD3 and MVPI in InvertedPendulum. Curves are averaged over 10 seeds with shaded regions indicating standard errors.

### 10.5 HalfCheetah

The agent controls a robot with two legs. The state dimension is 18 (add X-position). The action dimension is 6. The reward is determined by the speed between the current and the previous time step and a penalty over the magnitude of the input action (Originally, only speed toward right is positive, we make the speed positive in both direction so that agent is free to move left or right). If the agent reaches the region X-coordinate $< -3$, an additional noisy reward signal sampled from $\mathcal{N}(0, 1)$ times 10 is given. The game ends when a maximum episode length 500 is reached.

**Policy function.** The policy is a normal distribution in PPO, and PPO based methods (MVO, Tamar, MVP, and MG). The architecture is the same as in InvertedPendulum. Hidden size is 256.

The policy of TD3 and MVPI is the same as in InvertedPendulum. Hidden size is 256.

**Value function.** The value function in PPO, PPO based methods (MVO, Tamar, MVP, and MG), TD3 and MVPI is a neural network with two hidden layers. The hidden size is 256. Activation is ReLU.

**Optimizer.** Optimizer for policy and value is Adam.

### 10.5.1 Learning Parameters

Discount factor is $\gamma = 0.99$.

**Common parameters of PPO and PPO based methods.** GAE parameter: 0.95, Entropy coef: 0.01, Critic coef: 0.5, Clip $\epsilon$: 0.2, Grad norm: 0.5.

**PPO.** policy learning rate 3e-4 $\in$ {3e-4, 7e-5, 1e-5}, value function learning rate is the same. Inner update number $M = 5$.

**MVO.** policy learning rate 7e-5 $\in$ {3e-4, 7e-5, 1e-5}, value function learning rate is the same. Sample size $n = 10$. Inner update number 5.

**Tamar.** policy learning rate 7e-5 $\in$ {3e-4, 7e-5, 1e-5}, value function learning rate is the same. $J, V$ learning rate is 100 times policy learning rate. Threshold $b = 50 \in$ {10,50,100}. $\lambda = 0.2 \in$ {0.2, 0.4, 0.6}.

**MVP.** policy learning rate 7e-5 $\in$ {3e-4, 7e-5, 1e-5}, value function and $y$ learning rate is the same. $\lambda = 0.2 \in$ {0.2, 0.4, 0.6}.

**MG.** policy learning rate 3e-4 $\in$ {3e-4, 7e-5, 1e-5}, value function learning rate is the same. Sample size $n = 10$. Inner update number $M = 5$.

**TD3.** policy learning rate 3e-4 $\in$ {3e-4, 7e-5, 1e-5}, value function learning rate is the same. Batch size is 256.

**MVPI.** policy learning rate 3e-4 $\in$ {3e-4, 7e-5, 1e-5}, value function learning rate is the same. $\lambda = 0.2 \in$ {0.2, 0.4, 0.6}. Batch size is 256.

### 10.5.2 Return Variance and Gini Deviation in HalfCheetah

See Figures 14 and 15.

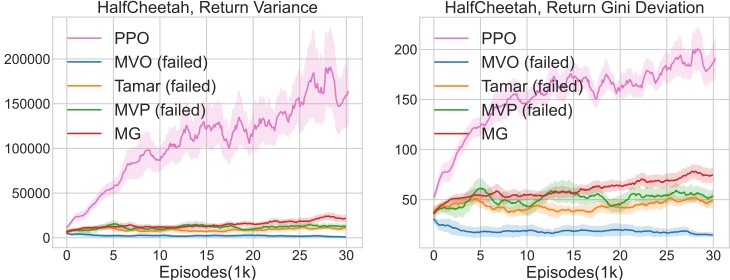

Figure 14: Return variance and Gini deviation of on-policy methods in HalfCheetah. Curves are averaged over 10 seeds with shaded regions indicating standard errors

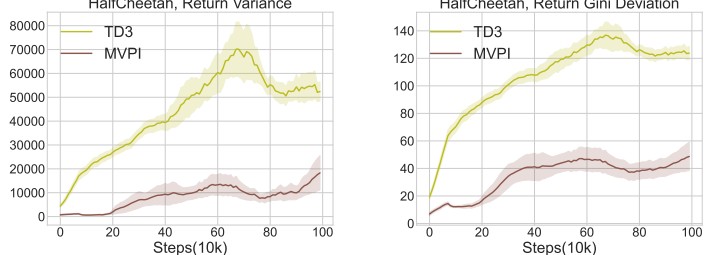

Figure 15: Return variance and Gini deviation of off-policy methods in HalfCheetah. Curves are averaged over 10 seeds with shaded regions indicating standard errors

### 10.6 Swimmer

The agent controls a robot with two rotors (connecting three segments) and learns how to move. The state dimension is 10 (add XY-positions). The action dimension is 2. The reward is determined by the speed between the current and the previous time step and a penalty over the magnitude of the input action (Originally, only speed toward right is positive, we make the speed positive in both direction so that agent is free to move left or right). If agent reaches the region X-coordinate $> 0.5$, an additional noisy reward signal sampled from $\mathcal{N}(0, 1)$ times 10 is given. The game ends when a maximum episode length 500 is reached.

The neural network architectures and learning parameters are the same as in HalfCheetah.

### 10.6.1 Return Variance and Gini Deviation in Swimmer

See Figures 16 and 17.

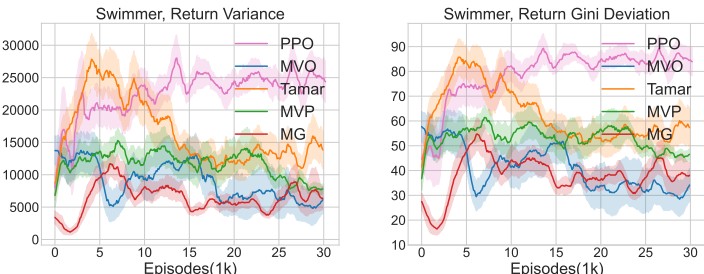

Figure 16: Return variance and Gini deviation of on-policy methods in Swimmer. Curves are averaged over 10 seeds with shaded regions indicating standard errors

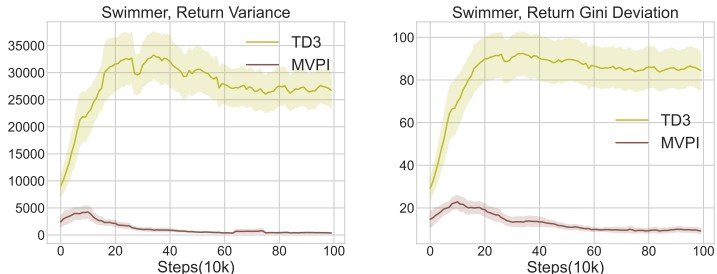

Figure 17: Return variance and Gini deviation of off-policy methods in Swimmer. Curves are averaged over 10 seeds with shaded regions indicating standard errors

## 11 Additional Results

We design two other senarios in HalfCheetah and Swimmer (marked as HalfCheetah1 and Swimmer1 in the figure's caption). Here the randomness of the noisy reward linearly decreases over the forward distance (right direction) the agent has covered. To encourage the agent to move forward, only the forward speed is positive. The additional noisy reward is sampled from $\mathcal{N}(0, 1)$ times 10 times $1 - \frac{X}{20}$ if $X$-position $> 0$. To maximize the expected return and minimize the risk, the agent should move forward as far as possible.

The learning parameters are the same as Section 10.5.

The distance agents covered in these two environments are shown in Figures 18 19.

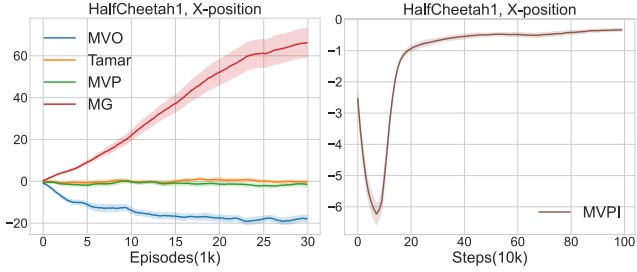

Figure 18: The distance agents covered in HalfCheetah1. Curves are averaged over 10 seeds with shaded regions indicating standard errors.

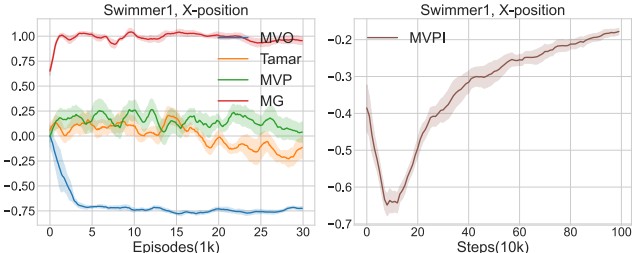

Figure 19: The distance agents covered in Swimmer1. Curves are averaged over 10 seeds with shaded regions indicating standard errors.

## 12 Additional Related Work and Discussion

### 12.1 Coherent measure of variability and risk

This paper focuses on the measure of *variability*, i.e., the dispersion of a random variable. Both variance and Gini deviation are measures of variability as discussed in Sections 2 and 3. Gini deviation is further known as a *coherent* measure of variability.

Another widely adopted risk measure is conditional value at risk (CVaR), which is a coherent *risk* measure [48]. Coherence is usually important in financial domains. Consider a continuous random variable $Z$ with the cumulative distribution function $F(z) = P(Z \leq z)$. The value at risk (VaR) or quantile at confidence level $\alpha \in (0, 1]$ is defined as $\text{VaR}_\alpha(Z) = \min\{z|F(z) \geq \alpha\}$. Then the CVaR at confidence level $\alpha$ is defined as

$$\text{CVaR}_\alpha(Z) = \frac{1}{\alpha} \int_0^\alpha \text{VaR}_\beta(Z) d\beta \tag{25}$$

Due to the nature of CVaR of only considering the tailed quantiles, it does not capture the variability of $Z$ beyond the quantile level $\alpha$. Thus it is not considered as a measure of variability. In [25], measures of variability and measures of risk are clearly distinguished and their coherent properties are different. We summarize their properties here [25].

Consider a measure $\rho : \mathcal{M} \to (-\infty, \infty]$

- (A) *Law-invariance*: if $X, Y \in \mathcal{M}$ have the same distributions, then $\rho(X) = \rho(Y)$
- (A1) *Positive homogeneity*: $\rho(\lambda X) = \lambda \rho(X)$ for all $\lambda > 0$, $X \in \mathcal{M}$
- (A2) *Sub-additivity*: $\rho(X + Y) \leq \rho(X) + \rho(Y)$ for $X, Y \in \mathcal{M}$
- (B1) *Monotonicity*: $\rho(X) \leq \rho(Y)$ when $X \leq Y$, $X, Y \in \mathcal{M}$
- (B2) *Translation invariance*: $\rho(X - m) = \rho(X) - m$ for all $m \in \mathbb{R}$, $X \in \mathcal{M}$
- (C1) *Standardization*: $\rho(m) = 0$ for all $m \in \mathbb{R}$
- (C2) *Location invariance*: $\rho(X - m) = \rho(X)$ for all $m \in \mathbb{R}$, $X \in \mathcal{M}$

**Coherent measure of variability**: A measure of variability is *coherent* if it satisfies properties (A), (A1), (A2), (C1) and (C2). An example is Gini deviation.

**Coherent risk measure**: A risk measure is *coherent* if it satisfies properties (A1), (A2), (B1) and (B2). An example is CVaR.

### 12.2 CVaR optimization in RL

CVaR focuses on the extreme values of a random variable and is usually adopted in RL domains to avoid catastrophic outcomes. Policy gradient is an important method to optimize CVaR, e.g., see [33][49][7]. Since CVaR focuses on the worst returns, policy gradient techniques often ignore high-return trajectories [33, 50]. [7] proposed cross entropy soft risk optimization to improve sample efficiency by optimizing with respect to all trajectories while maintaining risk aversion. A time difference method is proposed in [51]. CVaR is also used for action selection in distributional RL literature, e.g., see [35][52][53].