# OpenReview forum: "An Alternative to Variance: Gini Deviation for Risk-averse Policy Gradient"
_NeurIPS.cc/2023/Conference — NeurIPS 2023 poster_

### Official Review · Reviewer_Dz1b · 2023-07-03

**Soundness:** 2 fair
**Presentation:** 3 good
**Contribution:** 3 good
**Rating:** 5
**Confidence:** 3

**Summary:**

The paper points to limitations of the mean-variance criterion for risk-averse RL, and of the methods that optimize it. Instead, the paper proposes an alternative risk measure relying on Gini Deviation. The paper shows how to derive the PG for this metric, and demonstrates that it learns risk-averse policies more stably than methods that optimize the mean-variance.

**Strengths:**

1. The idea of using Gini Deviation for risk-averse RL is original as far as I know.
2. The metric is demonstrated as more stable than mean-variance.

**Weaknesses:**

Risk-aversion in RL is often handled by optimizing risk measures. Many risk measures have been studied, e.g., mean-variance, var, cvar, entropic, distortion measures (some of them are briefly mentioned in the paper). There are studies about both the properties of different measures and how to optimize them. In particular, commonly desired properties of the metric are captured by the notion of coherent risk measures. For example, mean-variance is not a coherent risk measure, but CVaR is, and many works have recently studied it.

A paper that proposes a new risk measure and claims to its benefits cannot ignore these works and the notion of coherence.

Considering the discussion above:
1. The presentation of the paper contribution lacks a lot of relevant context, hence the novelty and contribution are difficult to judge.
2. Relevant literature is hardly covered. There is no organized literature survey, and relevant works are not mentioned.
3. The experiments mostly demonstrate the weakness of the mean-variance risk measure, which is already known and cannot justify the suggested metric just by itself. Since the paper justification is essentially empirical, this is a quite critical issue.

Finally, the maze benchmark used in Sec 2.2, Sec 5.1 and Fig 1 is taken directly from [Greenberg et al. 2022], with neither crediting its original nor citing the corresponding work. Notice that using benchmarks and code (even if you modify them) without explicitly citing them in both paper and code may be considered Plagiarism.

**Questions:**

See the issues discussed above under Weaknesses.

Additional question:
As specified in the paper, the current method uses multiple trajectories for a single policy gradient. However, quantile regression (as in QRDQN) allows to learn quantile value representation without this limitation. Could such quantile regression be used to overcome the multi-trajectory limitation in mean-GD as well?

**Limitations:**

Algorithmic limitations are fairly discussed in Section 6.

---

> ### Author Rebuttal · Authors · 2023-08-10
>
> We thank the reviewer for the review. We address main concerns below.
>
> **For Weaknesses**
>
> 1. **Regarding the notion of coherence.**
>
> Thanks for pointing out the coherent risk measure. We are aware of this category of risk as mentioned in Section 1. However, as indicated by our title and abstract, we are aiming to address some limitations of variance related risk measures in this paper, instead of comparing with other coherent risk measures.
>
> Both variance and Gini deviation are measures of variability, while CVaR (a coherent risk measure) is a measure for extreme outcomes. They are targeting different application scenarios and users.
>
> Gini deviation is known as a **coherent measure of variability** (Line 161), e.g., see [22](Gini-type measures of risk and variability: Gini shortfall, capital allocations, and heavy-tailed risks). The properties of coherent measure of variability are different from the properties of coherent measure of risk, e.g., see Section 2.3 of [22].  We will briefly compare them in the revised version.
>
> 2. **Regarding lacks a lot of relevant context, and relevant literature is hardly covered.**
>
> Since our paper focuses on the variance related risk measures, i.e., variability. We give a detailed analysis of the main stream mean-variance methods in Section 2.
>
> 3. **The experiments mostly demonstrate the weakness of the mean-variance risk measure, which is already known and cannot justify the suggested metric just by itself.**
>
> Thanks for raising this concern. We think there may be a misunderstanding.  It is already known that variance is not a coherent risk measure, but this is not the weakness that we demonstrate nor try to resolve.  First we would like to invite the reviewer to read Section 2 of [22] (Gini-type measures of risk and variability: Gini shortfall, capital allocations, and heavy-tailed risks).  We focus on measures of variability (instead of measures of risk such as CVaR) as formally defined in [22].  Measures of variability are location invariant and consider the entire distribution of outcome instead of the tails.  They include variance and Gini-deviation, but not CVaR.  The confusion stems from the fact that the term "risk" is used casually in RL to refer to both measures of risk and variability.  However in this work we demonstrate that variance is not positive homogeneous (Line 161) and then propose to use Gini-deviation as a replacement since it is positive homogeneous and a coherent measure of variability (but not a coherent measure of risk).  We will add text to that effect in the paper.
>
> 4. **Regarding the missing citation of [Greenberg et al. 2022]**
>
> We acknowledge that the maze problem that we described in Figure 1 is a modified version of the Guarded Maze problem in [Greenberg et al. 2022] and that we failed to cite their work in our paper and our code.  We sincerely regret this omission.  While we had no intention of misappropriating their work, we now realize that this mistake may be seen as a form of plagiarism.  We are taking this seriously and we have resolved this issue directly with the authors.  The authors contacted us separately and independently of the NeurIPS reviews to point out the missing citation and work attribution.  We exchanged several emails with the authors and quickly fixed this mistake.  We ran by the authors a revised version of the manuscript where we did the following changes:
>
> 1) In Section 1, second paragraph, we added a citation to [Greenberg et al., 2022]
> 2) In Section 1, last paragraph, we changed "we created several domains" to "we modified several domains (Guarded Maze [7], Lunar Lander [15], Mujoco [16])".
> 3) In Figure 1: we indicated that the environment is a modified version of Guarded Maze and added a citation to [Greenberge el al., 2022]
> 4) In Section 5.1, we added the following explanation.  "The original Guarded Maze problem is asymmetric with two openings to reach the top path (in contrast to a single opening for the bottom path).  In addition, paths via the top tend to be longer than paths via the bottom. We modified the maze to be more symmetric in order to reduce preferences arising from certain exploration strategies that might be biased towards shorter paths or greater openings, which may confound risk aversion."
> 5) In the appendix, we added a discussion about additional related work in which we cited [Greenberg et al., 2022]
>
> After inspecting those changes, Greenberg wrote: "Thank you for your recent messages and for revising the manuscript. I am glad to see that our paper could help you in your very interesting work. The attribution in the revised paper that you sent seems entirely proper to me. I will be happy to see you in a conference."  Hence, if our paper is accepted, we will incorporate the changes summarized above that have been approved by Greenberg.
>
>
> **For Questions**
>
> 1. **Reagrding using QRDQN to learn quantile value representation**
>
> Thanks for this good question. QRDQN is a TD learning method for risk neutral decision making. Since most risk terms are not time consistent, i.e., optimizing risk at each time step is usually not the same as optimizing the total return risk at the initial state, incorporating quantile TD learning is thus challenging. We postpone this part to future work.

---

> > ### Comment · Reviewer_Dz1b · 2023-08-13
> >
> > I thank the authors and appreciate the detailed response.
> > I accept the responses, and increase my score accordingly.
> >
> > I believe that it will indeed be helpful to specify more clearly the notion of coherent variability and its importance.
> >
> > Regarding the literature: note that the title specify "risk-averse PG" as the objective, and note that very few of the cited related works are from the last 5 years. A paragraph about alternative risk-averse approaches may both help clarifying the context of this paper in the literature, and prevent confusion between different notions of coherence.

---

> > > ### Author Response · Authors · 2023-08-13
> > >
> > > Thanks for your reply and the suggestion! As mentioned in the rebuttal, we will definitely add more explanations to clarify variability and risk and discuss more related work in terms of other risk measures.

---

### Official Review · Reviewer_oAim · 2023-07-06

**Soundness:** 4 excellent
**Presentation:** 3 good
**Contribution:** 3 good
**Rating:** 6
**Confidence:** 4

**Summary:**

This paper considers the problem of risk-sensitive reinforcement learning in the policy optimization setting, under the mean-variance risk criteria. They provide background on the problem and demonstrate the issues with learning an optimal policy due to the estimation of the variance requiring double samples. They demonstrate that previous approaches to this problem which use the immediate reward variance as a proxy are not desirable (Section 2). They introduce the Gini deviation as a surrogate for the variance, and demonstrate that through a Choquet integral characterization, it can be estimated without the double-sampling issue by leveraging quantile distributional RL. They incorporate this into a PPO-style algorithm and compare its performance against others across a range of risk-sensitive benchmarks.

**Strengths:**

- Well presented motivation and contribution.
- Novel idea, which achieves strong empirical performance.

**Weaknesses:**

- There is little theoretical support for the algorithm, which would strengthen the contribution.

**Questions:**

- What is the space $\mathcal{M}^\infty$ introduced in Definition 1? It is never defined.

**Limitations:**

The authors discuss limitations of their work, and propose directions for future work.

---

> ### Author Rebuttal · Authors · 2023-08-10
>
> We thank the reviewer for the positive review. We address main concerns below.
>
> + **What is the space $M^\infty$ introduced in Definition 1?**
>
> We apologize for this typo. It should be $L^\infty$ to represent the set of bounded random variables, e.g., see original definition in Equation 1 of the paper [Characterization, Robustness and Aggregation of Signed Choquet Integrals].

---

> > ### Comment · Reviewer_oAim · 2023-08-12
> > **Response to rebuttal**
> >
> > I thank the authors for clarifying my confusion, and after reading the other reviews I am inclined to maintain my score.

---

### Official Review · Reviewer_cQhS · 2023-07-07

**Soundness:** 3 good
**Presentation:** 3 good
**Contribution:** 3 good
**Rating:** 7
**Confidence:** 3

**Summary:**

The paper proposes to restrict the Gini deviation of the return for risk-averse reinforcement learning (RARL). This is motivated by a discussion of the shortcomings of variance-based RARL methods, such as scale dependence and instability due to large squared returns. A derivation of an expression for the gradient of the Gini deviation of the return serves as a basis for a practical algorithm, mean-GD (MG). It adds a penalty proportional to the Gini deviation to the reinforcement learning objective. The distribution of returns, which is required for the gradient of the Gini deviation of the return, is approximated with a finite number of sampled trajectories. An empirical evaluation of MG in the tabular and function approximation settings shows advantages of MG in learning risk-averse policies over return and reward-based baselines.

**Strengths:**

The use of the Gini deviation as a new risk measure for RARL is motivated well by a concise discussion of the limitations of variance-based methods and is, to the best of my knowledge, new. The shortcomings of existing methods are illustrated with the help of a simple maze environment. The development of methods that are less sensitive to hyperparameters and environment properties such as reward scale is furthermore highly relevant for applications.

The derivation of a tractable expression for the gradient of the Gini deviation is done carefully and required assumptions are stated clearly. The experimental evaluation furthermore considers a sufficiently broad range of environments (tabular maze over lunar lander to MuJoCo environments) and baselines.

**Weaknesses:**

While the derivation of equations (15) and (16) are sufficiently clear, I would appreciate some intuition about the nature of the contribution of the Gini deviation penalty to the parameter update. The current version of the text does not really discuss the proposed algorithm but jumps right into the experimental evaluation. I think that for adoption of the algorithm by the community an intuitive understanding would be helpful.

There is furthermore a gap between the assumptions for the derivation and the actual environments used for the experiments and the illustration. The maze environment, for example, uses a shortest path objective which implies a discrete return. Many real-world environments will have complicated return distributions with atoms etc. It would be interesting to learn whether this poses a problem or if the assumptions are technical in nature and only required for the proof.

The significance of the experimental results is difficult to judge without further information. While I found the used hyperparameters in the supplementary material, I did not find information on how they were chosen. Was a hyperparameter optimization performed and, if yes, was it done manually, with a grid search or with some other algorithm? Without this information the frequent failures of the baselines are difficult to place. I would furthermore suggest to put vertical lines indicating the performance of MVPI in the figures (to avoid the problem of incompatible x axes). Right now it looks a bit like the most competitive baseline is only shown in the supplementary material where it is hard to compare it to the rest.

Related to this point: Part of the motivation for the proposed algorithm was to mitigate the need to extensively tune the hyperparameter $\lambda$ for different environments/reward functions. It would therefore be interesting to see how sensitive MG as well as the baselines are to the hyperparameters controlling the risk-averseness.

Typos:

* In line 42 “We show that our method can learn risk-averse policy […]”, there is an “a” missing.
* In line 69 the expression for the gradient of $M(\theta)$ contains an $a$ and a $u(\theta)$ which have not been introduced. Together with a different expression given in line 88, this looks like a typo.

**Questions:**

* It seems to me that the equality in line 234 is incorrect as the probability density of sampling a trajectory is not necessarily the same as the probability density of obtaining its reward. An extreme counter example would be an environment with a reward identical to zero but many possible trajectories. I do not think this poses a problem for the further derivation of MG as the set of trajectories that result in the same reward is independent of the policy parameters. However, it would still make sense to correct this formulation for better readability.
* As the policy is defined to output values in $[0, 1]$ (line 48), does this mean discrete actions are assumed?
* I would suggest to better relate the paragraph on the baselines (starting in line 281) to the discussion of prior methods in section 2. With the current formulation, it is not clear to the reader which baselines correspond to which of the methods discussed in section 2. Adding some clarification here would make interpreting the results a lot easier.
* I would be interested in learning why most of the environments were specifically designed for this paper. Are no standard environments for RARL available that would have been useful?
* Would it be possible to use the standard deviation instead of the variance to get scale invariance?
* Is the variance of the estimate for the CDF in equation (16) a problem?
* Related to the last question: Would it be possible to learn an approximation of the return distribution to reduce variance and get better convergence of the policy? The return distribution is clearly non-stationary but tracking non-stationary functions with estimates is not uncommon in reinforcement learning. Maybe this could help with asymptotic performance.
* Why is MG performing poorly on HalfCheetah while MVPI seems to do well?

**Limitations:**

The limitations of the MG algorithm, mainly low sample efficiency, have been addressed in a separate paragraph.

---

> ### Author Rebuttal · Authors · 2023-08-10
>
> We thank the reviewer for the positive review. We address main concerns below.
>
> **For Weaknesses**
>
> 1.**The current version does not really discuss the proposed algorithm but jumps right into the experiments.**
>
> The full algorithms are in Section 9 in appendix due to page limit.
>
> The procedures of the algorithms are 1. collect a group of trajectories 2. calculate the risk-neutral policy gradient 3. sort the trajectories by returns and calculate the Gini deviation policy gradient 4. update the policy by adding them together with a trade-off $\lambda$.
>
> 2.**Regarding the gap between the assumptions and the actual environments.**
>
> This gap can be easily closed since we can always add a truncated Gaussian noise to the reward to make the return continuous without changing the optimal solution. This assumption for continuity of the return variable is also common in the literature, e.g., see Section 2 of [Optimizing the CVaR via Sampling].
>
> 3.**Regarding the hyperparameters.**
>
> Since most of the comparing methods are also compared in the MVPI paper, e.g. Tamar, MVP, MVPI, TD3, we take the parameter searching range in MVPI as a reference and manually search by ourselves. We ensure each algorithm sweeps over the same number of hyperparameter combinations to make a fair comparison. We will add more detials in the revised version.
>
> 4.**Right now it looks a bit like the most competitive baseline is only shown in the supplementary material.**
>
> Thanks for raising this concern. MVPI is a TD learning method, i.e., update paremters at each environment step. Thus, it is not straightfoward to be compared with episode based methods. In addition, the MVPI plots were in appendix mainly due to the page limit. If the paper is accepted, we will move the figure to the main content using the extra page for accepted papers.
>
> 5.**Regading the sensitivity to lambda**
>
> We show the learning curves of total return based methods, e.g., MVO, Tamar, MVP, MG with different lambdas in the Maze domain in the uploaded PDF file.
>
> **For Questions**
>
> 1.**Regarding equation in line 234. An extreme counter example would be an environment with a reward identical to zero but many possible trajectories.**
>
> We think this one is closely related to the previous question on the continuous assumption of the return variable. The counter example here conflicts with this assumption.
>
> 2.**As the policy is defined in [0, 1](line 48), do you assume discrete actions?**
>
> We apologize for this typo. We do not assume the actions to be discrete. We will correct this in the next version.
>
> 3.**Regarding the environment design, and standard environments for RARL?**
>
> The reason for specific design is that the evaluations in previous mean-variance papers are either done (a) in small domains but without a clear understanding of why a particular type of algorithm can succeed or fail (we did this in Maze) or (b) in large domains but verifying the risk-aversion is not very straightforward (e.g., MVPI adds action noise to Mujoco. It reports the mean-variance score but it is unclear what  a risk averse policy is or whether risk-aversion is achieved). Thus we modify several domains where the risk-aversion can be clearly defined to better evaluate each algorithm.
>
> For standard environments for RARL, as far as we know, there is no common benchmark in all previous mean-variance papers in Section 2. For example, Tamar's paper used portofolio. MVP's paper used portofolio, american-style option, and optimal stopping. MVPI's paper used Mujoco with noisy action.
>
> 4.**Regarding using standard deviation to get scale invariance**
>
> First, tough Std is related to the signed Choquet integral, it is a supremum over some signed Choquet integrals, e.g., see Example 3 in the paper [Characterization, Robustness and Aggregation of Signed Choquet Integrals]. The gradient calculation is in general hard.
>
> Second, directly computing the gradient of Std is possible. $Std = Var^{\frac{1}{2}}$, then $\nabla Std = \frac{1}{2}\frac{1}{\sqrt{Var}}\nabla Var$. This gradient may still suffer from numerical scale issues due to the $\nabla Var$ term.
>
> 5.**Regarding the variance of the estimator**
> To get an intuitive understanding of the variance, we can compare our updating rule with REINFORCE. In REINFORCE, the return term is multiplied to the sum of log pi.
>
> The variance of $\eta_i$ depends on the variance of ordered samples, which may not have a closed form. However, compared with the return in REINFORCE, $\eta_i$ is calculated based on the difference between ordered returns times a value in (0,1), which has a much smaller numerical scale, thus smaller variability.
>
> 6.**Would it be possible to learn an approximation of the return distribution to reduce variance and get better convergence?**
>
> Thanks for this suggestion. Learning a return distribution function can leverage distributional RL. However, due to the non time consistency of the risk term, incorporating TD method (distributional RL uses TD learning) is challenging, since optimizing risk at each time step is usually not the same as optimizing the total return risk at the initial state. We postpone this part as our future work.
>
> 7.**Why is MG performing poorly on HalfCheetah while MVPI seems to do well?**
>
> We designed two HalfCheetah domains (the second in Appendix).
>
> One is adding a noisy reward based on X-location (X<-3). In this domain, MVPI performs good since (a) MVPI is built on top of TD3. MG is built on top of PPO. TD3 learns faster and better than PPO in this domain as shown in Figure 5 of TD3 paper [Addressing Function Approximation Error in Actor-Critic Methods]. (b) MVPI's reward modification strategy naturally prevents it from visiting the X<-3 region.
>
> The other is adding linearly decayed noisy reward based on the distance the agent has covered. The distance that agents covered is in Figure 19 (Section 11). It is clear that MVPI gets stuck in this domain, i.e., final X location is close to the origin.

---

> > ### Comment · Reviewer_cQhS · 2023-08-14
> > **Response to rebuttal**
> >
> > I thank the authors for the detailed response.
> >
> > I appreciate their explanations on how to close the gap between Assumption 1 to 3 and the benchmarking environments, on the choice of benchmarking environments, possible downsides of substituting the standard deviation for the Gini deviation, the variance of the estimator for the GD gradient, and the challenges associated to leveraging distributional RL for RARL, and the performance of MVPI on two versions of the HalfCheetah environment.
> >
> > It is a useful addition to the paper that the authors will discuss the process of hyperparameter tuning in the revised paper, and will integrate the MVPI results better in the main text.  I would encourage them to also include the new experiments on the sensitivity with respect to the lambda parameter.
> >
> > Regarding equation in line 234, I thank the authors for pointing out that the counter example I gave violates Assumption 1. The example was intended to illustrate that, in general, the probability (density) of obtaining a specific trajectory is not the same as the probability (density) of obtaining it’s return as multiple trajectories could lead to the same return. For this reason line 234 appears imprecise to me. This is only a minor point, however, and does not affect the validity of the results. I would leave it to the authors to decide whether to change this formulation.
> >
> > I furthermore appreciate that the issue of a missing citation, pointed out by another reviewer, was resolved.
> >
> > I consider my questions and concerns sufficiently addressed, thank you.

---

> > > ### Author Response · Authors · 2023-08-15
> > >
> > > Thanks for your reply and the suggestion! We will add more explanation to Line 234 for better readability.

---

> > > > ### Comment · Reviewer_cQhS · 2023-08-15
> > > >
> > > > Thank you, I appreciate that!

---

### Official Review · Reviewer_FmwV · 2023-07-10

**Soundness:** 4 excellent
**Presentation:** 4 excellent
**Contribution:** 4 excellent
**Rating:** 7
**Confidence:** 3

**Summary:**

This paper presents a new approach to measure the risk for risk-averse reinforcement learning problems.
To be more precise, traditional methods usually use the variance of total return or per-step reward as the measurement of risk, since it is known that the variance is easy to calculate. Though easy to calculate, variance-based methods are observed to be sensitive to numerical scale and hard to optimize the policy. Based on the observation, the authors propose to use Gini deviation as the alternative to variance for measuring the risk. The authors study the properties of the new measurement and present how to estimate Gini deviation using empirical samples. Empirical experiments demonstrate that the new proposed methods can mitigate the limitation of variance-based methods while it can still learn high reward returns, compared with previous methods.

**Strengths:**

- The paper is well written, and the authors did an excellent job of presenting the limitation of variance methods before introducing Gini Deviation. As a result, readers can get a sense of the motivation for using Gini deviation as an alternative.

- The authors thoroughly introduce the properties of Gini deviation and present how to estimate this quantity using empirical samples. Besides the authors also introduce how to estimate this quantity under the setting of using importance sampling and PPO. And I think through this, authors can get a better understanding how to use the quantity in different variant of policy gradient methods.

- The authors also demonstrate the advantage of Gini Derivation via empirical experiments, for both tabular, discrete, and continuous control experiments.



**Weaknesses:**

- Though the authors present that Gini Deviation can be estimated via empirical samples, my primary concern is that it does not have close form and thus its estimation totally depends on the quality of the samples, if we do not have enough samples, then the variance or the bias of the estimation can lead to undesired optimization behavior.

- Since Gini Deivation requires estimating the quantile, it may introduce additional complexity compared with variance-based method.

- The authors claim that variance-based methods might be sensitive to numerical scales, but I have not seen any ablation study to directly compare this with Gini Deivation-based policy gradient method.

**Questions:**

- Can you clarify if we use empirical samples to estimate Gini Deviation, what would be the draw back? For example, what's the variance and bias of the new estimator using empirical samples if it has?

- Would you mind explaining the complexity of the new proposed method?

- Also, is it possible to demonstrate that with the new proposed measurement, the new estimator would be robust to numerical scales? (Maybe no need for empirical experiments, some intuition explanation would also be okay).

---

> ### Author Rebuttal · Authors · 2023-08-10
>
> We thank the reviewer for the positive review. We find the questions in "Weaknesses" and "Questions" are closely related. We combine the related ones and address main concerns below.
>
> 1. **Regarding the draw back, bias and variance of the sample based estimator**
>
> As discussed in the limitation section (section 6), the draw back of our sample based estimator is that we need a sufficient number of samples for each update due to estimating the CDF. However, we would also like to highlight that using multiple trajectory samples to estimate CDF or inverse CDF is also common in static CVaR related policy gradient methods, e.g., see papers [Optimizing the CVaR via sampling; Efficient risk-averse reinforcement learning].
>
> Generally, rigorous characterizing the bias and variance of $\eta_i$ in Equation 16 requires the true CDF, which is unknown. Also the bias and variance of $\eta_i$ involves the bias and variance of ordered samples, which are also complex. We give an intuitive analysis first and then demonstrate by simutation.
>
> The bias of $\eta_i$ comes from the difference between the empirical CDF and the true CDF. This bias will decrease when the sample size increases as the empirical CDF will be closer to the true CDF.
>
> For variance, we can compare our updating rule with REINFORCE. In REINFORCE, the return term is multiplied to the sum of log pi, while in GD policy gradient, $\eta_i$ is multiplied to the sum of log pi. Compared with this return term, $\eta_i$ is calculated based on the difference between ordered returns times a value in (0,1), which has a much smaller numerical scale, thus smaller variability.
>
> We demonstrate the bias and variance by simulating from a truncated Gaussian. Suppose ordered samples $x_{(1)}, x_{(2)},...,x_{(n)}$ (i.e., the samples are sorted in increasing order) are from a truncated Gaussian in range [-25, 25] with mean 0 and variance $10^2$. For $\eta_i=\sum_{i=1}^{n-1} \frac{2i}{n} (x_{(i+1)} - x_{(i)})-(x_{(n)}-x_{(i)})$, the bias is $\delta_i = \eta_i -  \int_{x_{(i)}}^{25} 2F(t)dt + (25 - x_{(i)})$.
>
> We can estimate the bias as $B=\mathbb{E}[\frac{1}{n-1}\sum_{i=1}^{n-1}\delta_i]$. When $n=10$, $B\approx -9.32$. When $n=50$, $B\approx -4.53$. When $n=100$, $B\approx -2.97$. The bias decreases as the sample size increases.
>
> We can estimate the variance as $V=\mathbb{E}[\frac{1}{n-1}\sum_{i=1}^{n-1}(\eta_i - \bar{\eta})^2]$, where $\bar{\eta}=\frac{1}{n-1}\sum_{i=1}^{n-1}\eta_i$. When $n=10$, $V\approx 6.96$. When $n=50$, $V\approx 10.79$. When $n=100$, $V\approx 11.66$. This variance is much smaller than the variance of the sampling distribution, i.e, 100. Here we can treat $x_{(i)}$ as the return term used in REINFORCE and $\eta_i$ be the term in GD gradeint. This indicates the variance of $\eta_i$ in GD gradient is relatively small.
>
> 2. **Regarding the complexity of the new proposed method?**
>
> Since we are updating gradients based on episodes, we suppose that computing the gradient for one trajectory is one time unit. Suppose the total number of training trajectories is $N$. Consider the mean-GD gradient built on top of PPO, based on importance sampling, the gradient for each trajectory is updated a fixed number of times $m$. Then the total complexity is $mN$ time units.
>
> When mean-GD gradient is built on top of the REINFORCE baseline, the complexity is in general between $N$ to $mN$ time units since the importance sampling update is terminated if the ratio is too extreme.
>
> 3. **Regarding the intuition of GD be more robust to numerical scales than varaince**
>
> Thanks for this good question. The intuition directly comes from the definitions of Gini deviation and variance as shown in Equation 6 and in Line 149. Thus $\mathbb{V}[cX]=c^2\mathbb{V}[X]$, while $\mathbb{D}[cX]=c\mathbb{D}[X]$ for $c>0$. We also highlight this property in Line 159. As a result, scaling the reward or the return will scale the Gini deviation linearly, but quadratically for variance.
>
> 4. **Regarding compare GD and variance with respect to the sensitivity to numerical scales.**
>
> We directly compare mean-variance with mean-Gini deviation in Section 5.1.  Following Line 299 in the paper, i.e. the sentence "The failure of variance-based baselines under simple reward manipulation", we provide an analysis of our experiments by modifying the numerical scale of the goal reward of the maze environment. This simple manipulation affects the mean-variance methods, but makes little difference to mean-Gini deviation as shown in Figure 2.

---

> > ### Comment · Reviewer_FmwV · 2023-08-20
> > **Response**
> >
> > Thank the authors for answering my questions.  I will keep my score and vote for acceptance. Please incorporate related discussion in your next version.

---

### Official Review · Reviewer_diaH · 2023-07-11

**Soundness:** 3 good
**Presentation:** 3 good
**Contribution:** 3 good
**Rating:** 5
**Confidence:** 2

**Summary:**

This paper discusses the challenges of mean-variance RARL methods, and proposes the Gini deviation as a substitute to variance penalization.

**Strengths:**

The paper is clearly written and well-motivated (by the disadvantages of return and instantaneous reward variance). The proposed method is evaluated in several experiments with comparator baselines, which seem convincing.

**Weaknesses:**

It seems the proposed algorithm has to sort its samples in every iteration (L245-248) which seems computationally expensive, and which may be sensitive to the parameters chosen for $n$?

While I recognize that this is a largely empirical paper, I suppose the paper could have been made more complete by providing some convergence guarantees for policy gradient, e.g., to a stationary point.

**Questions:**

Included in “weaknesses”.

**Limitations:**

yes

---

> ### Author Rebuttal · Authors · 2023-08-10
>
> We thank the reviewer for the positive review. We address main concerns below.
>
> + **It seems the proposed algorithm has to sort its samples in every iteration (L245-248) which seems computationally expensive, and which may be sensitive to the parameters chosen for $n$?**
>
> The sample here corresponds to trajectory's return. In practice, sorting samples in our experiments is in general not very time consuming since the maximum number of trajectory's return to sort is only 50 (in the maze problem in Section 5.1).
>
> Regarding the sensitivity, we did not find it difficult to choose $n$ in the experiments. For instance, $n$=10 in HalfCheetah and Swimmer (in Section 5.3) is reasonably good to achieve a low chance of visiting the high variance region in our experiments.

---

### Author Rebuttal · Authors · 2023-08-10

We include a PDF file which contains additional results requested by reviewer cQhS, regarding the sensitivity to $\lambda$

---

### Decision · Program_Chairs · 2023-09-21

**Decision:**

Accept (poster)

**Comment:**

The authors investigated the application of the Gini-deviation (equation 6) risk measure in risk-averse Reinforcement Learning (RL). To do so, additional objective was added to the standard average cumulative reward objective of the RL problem. Further, a policy gradient based algorithm, which minimizes the new objective (equation 9), was implemented and tested. By conducting a thorough experimental study the advantages of the Gini-deviation objective was highlighted in the context of RL, compared to other previously suggested risk-averse measures.

The reviewers were in a consensus for the acceptance of this paper due to the novelty of combining the Gini-deviation risk measure in RL, the derivation of a policy gradient based algorithm for the objective, and the empirical study the authors conducted. Constructive feedback, given by the reviewers during the discussion period, will be combined in the final version of the paper. For these reasons I am pleased to support the acceptance of this work.